# OCTOPACK: INSTRUCTION TUNING CODE LARGE LANGUAGE MODELS

**Niklas Muennighoff**    **Qian Liu**    **Armel Zebaze**    **Qinkai Zheng**    **Binyuan Hui**

**Terry Yue Zhuo**    **Swayam Singh**    **Xiangru Tang**    **Leandro von Werra**    **Shayne Longpre**

n.muennighoff@gmail.com

## ABSTRACT

Finetuning large language models (LLMs) on instructions leads to vast performance improvements on natural language tasks. We apply instruction tuning using code, leveraging the natural structure of Git commits, which pair code changes with human instructions. We compile COMMITPACK: 4 terabytes of Git commits across 350 programming languages. We benchmark COMMITPACK against other natural and synthetic code instructions (xP3x, Self-Instruct, OASST) on the 16B parameter StarCoder model, and achieve state-of-the-art performance among models not trained on OpenAI outputs, on the HumanEval Python benchmark (46.2% pass@1). We further introduce HUMANEVALPACK, expanding the HumanEval benchmark to a total of 3 coding tasks (Code Repair, Code Explanation, Code Synthesis) across 6 languages (Python, JavaScript, Java, Go, C++, Rust). Our models, OCTOCODER and OCTOGEEX, achieve the best performance across HUMANEVALPACK among all permissive models, demonstrating COMMITPACK's benefits in generalizing to a wider set of languages and natural coding tasks. Code, models and data are freely available at https://github.com/bigcode-project/octopack.

## 1) CommitPack

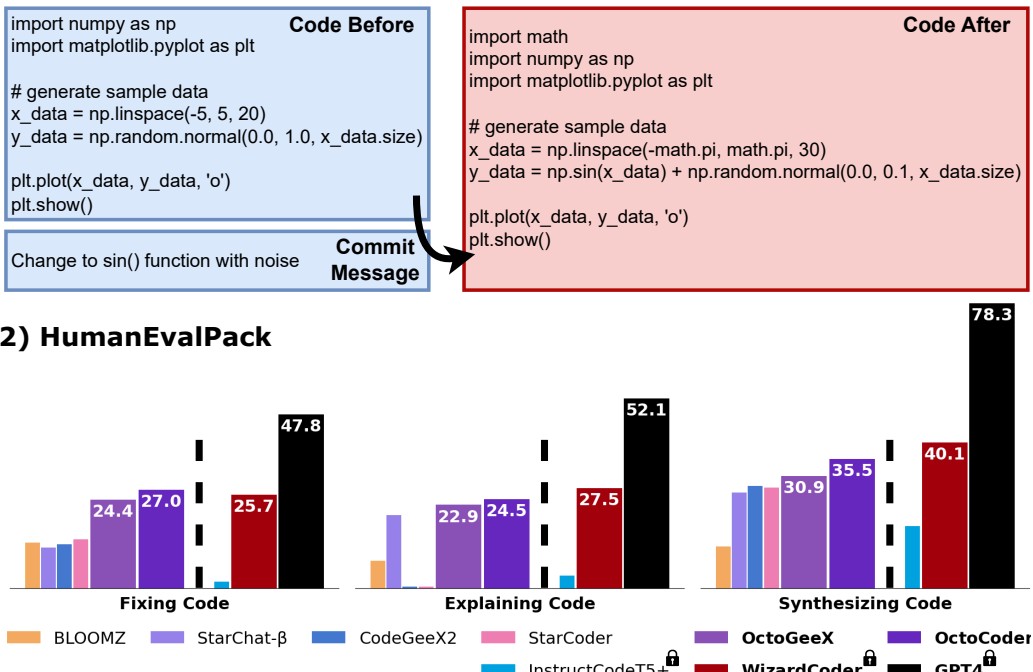

## 2) HumanEvalPack

Figure 1: **OCTOPACK Overview.** *1)* Sample from our 4TB dataset, COMMITPACK. *2)* Performance of OCTOCODER, OCTOGEEX and other code models including non-permissive ones (WizardCoder, GPT-4) on HUMANEVALPACK spanning 3 coding tasks and 6 programming languages.

## 1 INTRODUCTION

Finetuning large language models (LLMs) on a variety of language tasks explained via instructions (instruction tuning) has been shown to improve model usability and general performance (Wei et al., 2022; Sanh et al., 2022; Min et al., 2022; Ouyang et al., 2022). The instruction tuning paradigm has also proven successful for models trained on visual (Liu et al., 2023a; Li et al., 2023a), audio (Zhang et al., 2023b) and multilingual (Muennighoff et al., 2022b; Wang et al., 2022b) data.

In this work, we instruction tune LLMs on the coding modality. While Code LLMs can already be indirectly instructed to generate desired code using code comments, this procedure is brittle and does not work when the desired output is natural language, such as explaining code. Explicit instructing tuning of Code LLMs may improve their steerability and enable their application to more tasks. Concurrently to our work, three instruction tuned Code LLMs have been proposed: PanGu-Coder2 (Shen et al., 2023), WizardCoder (Luo et al., 2023) and InstructCodeT5+ (Wang et al., 2023c). These models rely on more capable and closed models from the OpenAI API[1] to create their instruction training data. This approach is problematic as **(1)** closed-source APIs keep changing and have unpredictable availability (Pozzobon et al., 2023; Chen et al., 2023a), **(2)** it relies on the assumption that a more capable model exists **(3)** it can reinforce model hallucination (Gudibande et al., 2023) and **(4)**, depending on legal interpretation, OpenAI's terms of use[2] forbid such models: "...You may not...use output from the Services to develop models that compete with OpenAI...". Thus, we consider models trained on OpenAI outputs not usable for commercial purposes in practice and classify them as non-permissive in this work.

We focus on more permissively licensed data and avoid using a closed-source model to generate synthetic data. We benchmark four popular sources of code instruction data: **(1)** xP3x (Muennighoff et al., 2022b), which contains data from common code benchmarks, **(2)** Self-Instruct (Wang et al., 2023a) data we create using a permissive Code LLM, **(3)** OASST (Köpf et al., 2023), which contains mostly natural language data and few code examples and **(4)** COMMITPACK, our new 4TB dataset of Git commits. Instruction tuning's primary purpose is to expand models' generalization abilities to a wide variety of tasks and settings. Thus, we extend the code synthesis benchmark, HumanEval (Chen et al., 2021; Zheng et al., 2023), to create HUMANEVALPACK: A code benchmark covering code synthesis, code repair, and code explanation across six programming languages.

Instruction tuning StarCoder (Li et al., 2023b) on a filtered variant of COMMITPACK and OASST leads to our best model, OCTOCODER, which surpasses all other openly licensed models (Figure 1), but falls short of the much larger GPT-4 (OpenAI, 2023). GPT-4 is close to maximum performance on the code synthesis variant, notably with a pass@1 score of 86.6% on Python HumanEval. However, it performs significantly worse on the code fixing and explanation variants of HUMANEVALPACK, which we introduce. This suggests that the original HumanEval benchmark may soon cease to be useful due to models reaching close to the maximum performance. Our more challenging evaluation variants provide room for future LLMs to improve on the performance of the current state-of-the-art.

In summary, we contribute:

- COMMITPACK and COMMITPACKFT: 4TB of permissively licensed code commits across 350 programming languages for pretraining and a filtered 2GB variant containing high-quality code instructions used for finetuning
- HUMANEVALPACK: A benchmark for Code LLM generalization, spanning three scenarios (Code Repair, Code Explanation, Code Synthesis) and 6 programming languages (Python, JavaScript, Java, Go, C++, Rust)
- OCTOCODER and OCTOGEEX: The best permissive Code LLMs

## 2 COMMITPACK: CODE INSTRUCTION DATA

Prior work has shown that models can generalize to languages included in pretraining, but absent during instruction tuning (Muennighoff et al., 2022b). However, they also show that including such

---

[1]https://openai.com/blog/openai-api
[2]https://openai.com/policies/terms-of-use

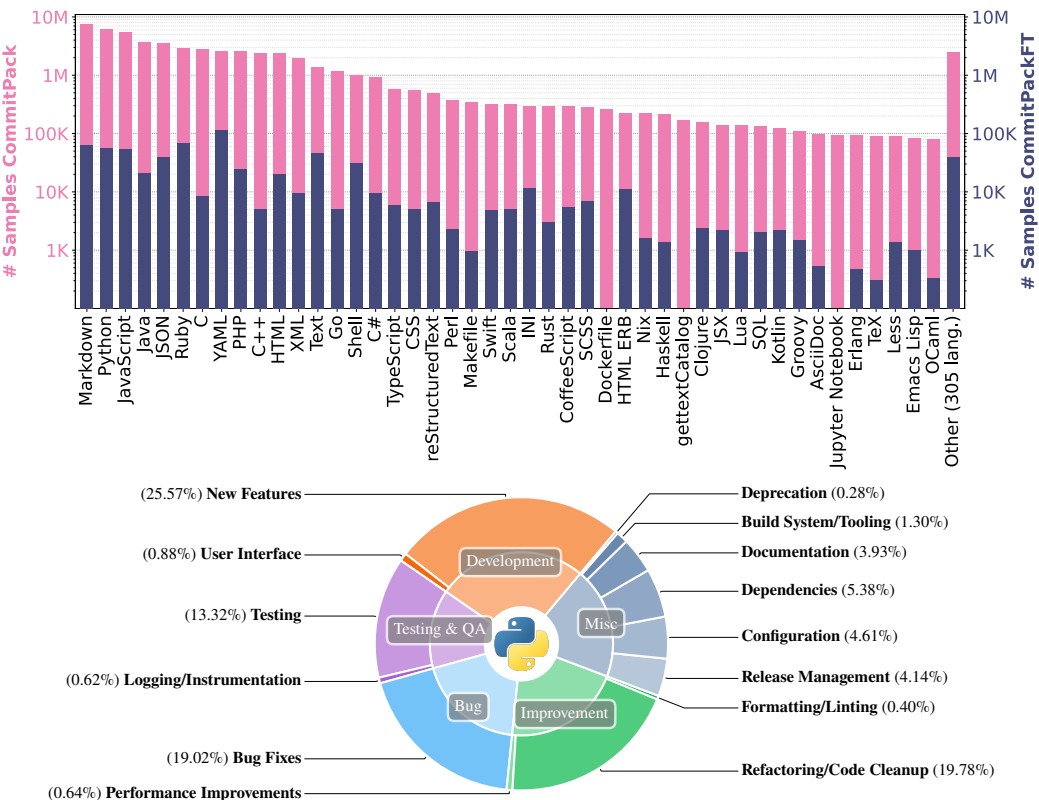

Figure 2: **Overview of COMMITPACK and COMMITPACKFT.** *Top:* Language distribution of the full commit data (COMMITPACK) and the variant filtered for high-quality instructions (COMMITPACKFT). See Appendix C for the full distribution. *Bottom:* Task distribution of commits on the Python subset of COMMITPACKFT (59K samples) according to GPT-4.

| Dataset (↓) | Base dataset | | | Subset | | |
|---|---|---|---|---|---|---|
| | Lang. | Samples | Code fraction | Lang. | Samples | Code fraction |
| xP3x | 8 | 532,107,156 | 0.67% | 8 | 5,000 | 100% |
| StarCoder Self-Instruct | 12 | 5,003 | 100% | 12 | 5,003 | 100% |
| OASST | 49 | 161,443 | 0.9% | 28 | 8,587 | 2.5% |
| COMMITPACKFT | 277 | 742,273 | 100% | 6 | 5,000 | 100% |

Table 1: **Statistics of code instruction data we consider.** We display the number of programming languages, total samples, and fraction of samples that contain code for permissive instruction datasets. For finetuning on these datasets, we use small subsets with around 5,000 samples each.

languages during instruction tuning boosts their performance further. We hypothesize that code data exhibits the same behavior. To improve performance on code-related tasks, we thus construct a code instruction dataset leveraging the natural structure of Git commits.

**COMMITPACK**   To create the dataset, we use commit metadata from the GitHub action dump on Google BigQuery.[3] We apply quality filters, filter for commercially friendly licenses, and discard commits that affect more than a single file to ensure commit messages are very specific and to avoid additional complexity from dealing with multiple files. We use the filtered metadata to scrape the affected code files prior to and after the commit from GitHub. This leads to almost 4 terabytes of data covering 350 programming languages (COMMITPACK). As instruction tuning does not require so much data (Zhou et al., 2023a; Touvron et al., 2023), we apply several strict filters to

---

[3]https://www.gharchive.org/

reduce the dataset to 2 gigabytes and 277 languages (COMMITPACKFT). These include filtering for samples where the commit message has specific words in uppercase imperative form at the start (e.g. "Verify ..."), consists of multiple words, and does not contain external references. All filters are detailed in Appendix D. Figure 2 depicts the distribution of both datasets and the tasks contained in COMMITPACKFT. For instruction tuning our models, we select 5,000 random samples from COMMITPACKFT across the 6 programming languages that we evaluate on. In Appendix G, we also experiment with pretraining on the entirety of COMMITPACK.

**Alternatives**   We consider three additional datasets for instruction tuning presented in Table 1. **xP3x**: xP3x is a large-scale collection of multilingual instruction data with around 532 million samples (Muennighoff et al., 2022b). We focus only on the code subset of xP3x, excluding Neural-CodeSearch (Li et al., 2019) which is not licensed permissively, and select 5,000 samples. **Self-Instruct**: Using the Self-Instruct method (Wang et al., 2022a) and the StarCoder model (Li et al., 2023b), we create 5,003 synthetic instructions and corresponding answers. **OASST**: OASST is a diverse dataset of multi-turn chat dialogues (Köpf et al., 2023). Only a few of the dialogues contain code. We reuse a filtered variant from prior work (Dettmers et al., 2023) and additionally filter out moralizing assistant answers (Appendix D) leading to 8,587 samples.

# 3   HUMANEVALPACK: EVALUATING INSTRUCTION TUNED CODE MODELS

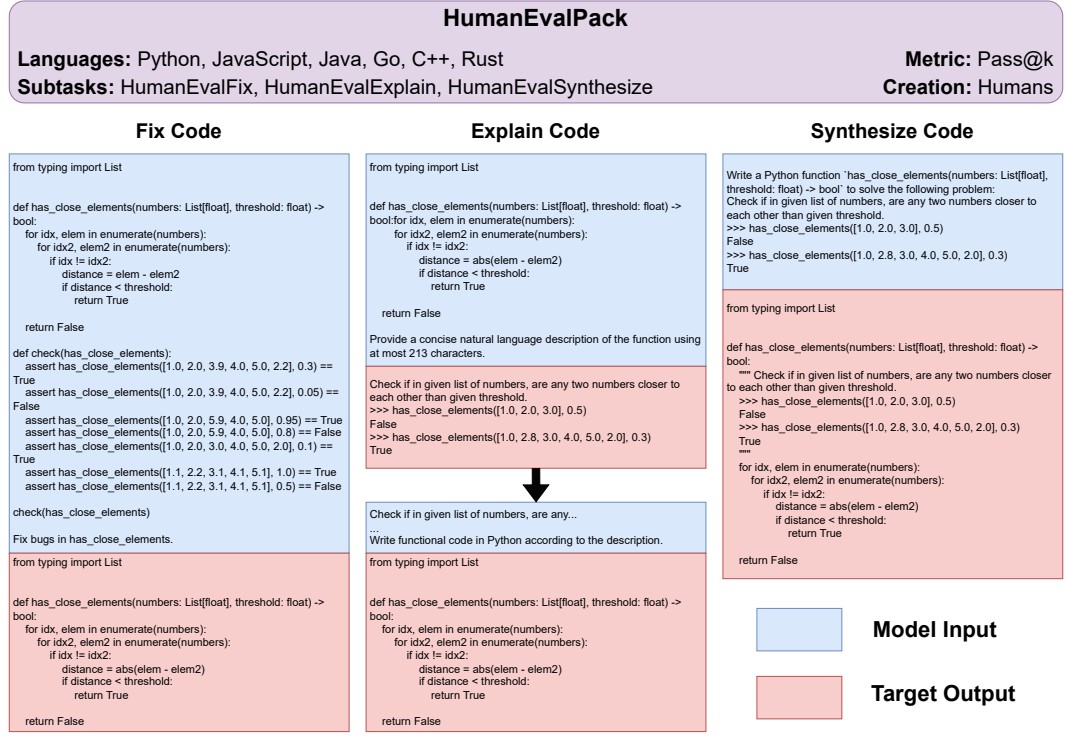

Figure 3: **HUMANEVALPACK overview.** The first HumanEval problem is depicted across the three scenarios for Python. The bug for HUMANEVALFIX consists of a missing "abs" statement.

When instruction tuning LLMs using natural language (NL) data, the input is an NL instruction with optional NL context and the target output is the NL answer to the task (Wei et al., 2022). When instruction tuning with code (C) data, code may either appear *only in the input* alongside the NL instruction (NL+C→NL, e.g. code explanation), *only in the output* (NL→C, e.g. code synthesis), or in *both input and output* (NL+C→C, e.g. code modifications like bug fixing). While prior benchmarks commonly only cover variants of code synthesis, users may want to use models in all three scenarios. Thus, we expand the code synthesis benchmark HumanEval (Chen et al., 2021; Zheng et al., 2023) to cover all three input-output combinations for six languages (Figure 3).

**HUMANEVALFIX (NL+C→C)**   Given an incorrect code function with a subtle bug and accompanying unit tests, the model is tasked to fix the function. We manually add a bug to each of the 164 HumanEval solutions across all 6 languages (984 total bugs). For a given sample, the bugs are as similar as possible across the 6 languages enabling meaningful comparison of scores across languages. Bugs are written such that the code still runs but produces an incorrect result leading to at least one unit test failing. Bug statistics and examples are in Appendix L. We also evaluate an easier variant of this task where instead of unit tests, models are provided with the correct function docstring as the source of truth to fix bugs, see Appendix K.

**HUMANEVALEXPLAIN (NL+C→NL)**   Given a correct code function, the model is tasked to generate an explanation of the code. Subsequently, the same model is tasked to regenerate the code given only its own explanation. The second step allows us to score this task via code execution and measure pass@$k$ (Chen et al., 2021) instead of evaluating the explanation itself using heuristic-based metrics like BLEU (Papineni et al., 2002) or ROUGE (Lin, 2004) which have major limitations (Reiter, 2018; Schluter, 2017; Eghbali & Pradel, 2022; Zhou et al., 2023b). To prevent models from copying the solution into the description, we remove any solution overlap of at least 20 characters from the description. We further enforce a character length limit on the model-generated explanation equivalent to the length of the docstring describing the function. This limit is specified in the prompt for the model. Note that the function docstring itself is never provided to the model for this task.

**HUMANEVALSYNTHESIZE (NL→C)**   Given a natural language docstring or comment describing the desired code, the model is tasked to synthesize the correct code. This task corresponds to the original HumanEval benchmark (Chen et al., 2021). For instruction tuned models, we add an explicit instruction to the input explaining what the model should do. For models that have only gone through language model pretraining, we follow Chen et al. (2021) and provide the model with the function header and docstring to evaluate its completion of the function.

For all tasks we execute the code generations to compute performance using the pass@$k$ metric (Chen et al., 2021): a problem is considered solved if any of $k$ code generations passes every test case. We focus on the simplest version of pass@$k$, which is pass@1: the likelihood that the model solves a problem in a single attempt. Like Chen et al. (2021), we use a sampling temperature of 0.2 and $top_p = 0.95$ to estimate pass@1. We generate $n = 20$ samples, which is enough to get reliable pass@1 estimates (Li et al., 2023b). For GPT-4, we generate $n = 1$ samples. Using $n = 1$ instead of $n = 20$ for GPT-4 only changed scores from 75.0% to 75.2% pass@1 on HUMANEVALSYNTHESIZE Python while providing 20x cost savings.

Python HumanEval is the most widely used code benchmark and many training datasets have already been decontaminated for it (Kocetkov et al., 2022). By manually extending HumanEval, we ensure existing decontamination remains valid to enable fair evaluation. However, this may not hold for all models (e.g. GPT-4), thus results should be interpreted carefully.

## 4   OCTOCODER: BEST COMMERCIALLY LICENSED CODE LLM

### 4.1   ABLATING INSTRUCTION DATA CHOICES

We instruction tune the pretrained StarCoder model (Li et al., 2023b) on different combinations of our instruction datasets (§2). We evaluate all models on the Python subset of HUMANEVALPACK as depicted in Figure 4. Similar to prior work (Taori et al., 2023), we format all instructions into a consistent schema to distinguish question and answer (see Figure 18).

**COMMITPACKFT enables CodeLLMs to fix bugs**   COMMITPACKFT is critical for the performance boost on code repair (HUMANEVALFIX), where instruction tuning on only OASST or other variants results in a significantly lower score. This is likely due to COMMITPACKFT including around 20% of bug fixes among other code-related tasks (Figure 2).

**Importance of samples with natural language targets**   The pretrained StarCoder model, as well as the Self-Instruct variant, perform poorly on code explanation (HUMANEVALEXPLAIN). This is because both models are only conditioned to write code instead of natural language. We find that to

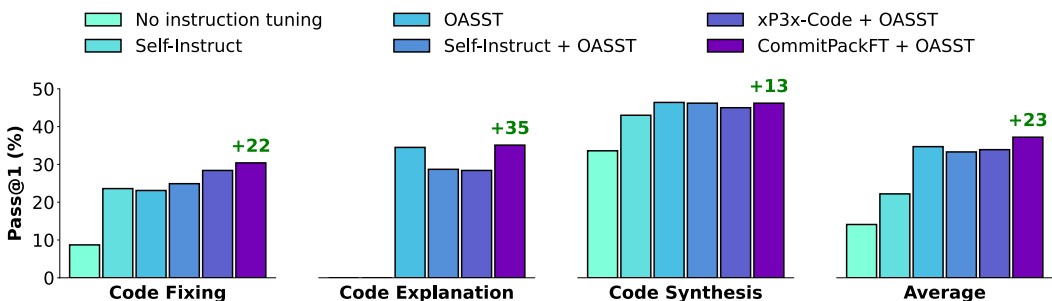

Figure 4: **Comparing permissively licensed instruction datasets by instruction tuning StarCoder.** Models are evaluated on the Python subset of HUMANEVALPACK.

perform well at explaining code, it is necessary to include samples with natural language as the target output during instruction tuning. Only relying on data with code as the target, such as the Self-Instruct data, will lead to models always outputting code even if the question requires a natural language output. Thus, we mix all other ablations with OASST, which contains many natural language targets. While the xP3x subset also contains samples with natural language output, many of its target outputs are short, which leads to models with a bias for short answers. This is impractical for the explanation task leading to the comparatively low score of mixing xP3x with OASST.

**COMMITPACKFT+OASST yields best performance**   All instruction datasets provide similar boosts for code synthesis (HUMANEVALSYNTHESIZE), which has been the focus of all prior work on code instruction models (Wang et al., 2023c; Luo et al., 2023; Muennighoff et al., 2022b). We achieve the best average score by instruction tuning on COMMITPACKFT mixed with our filtered OASST data yielding an absolute 23% improvement over StarCoder. Thus, we select COMMITPACKFT+OASST for our final model dubbed OCTOCODER. Using the same data, we also instruction tune the 6 billion parameter CodeGeeX2 (Zheng et al., 2023) to create OCTOGEEX. Training hyperparameters for both models are in Appendix P.

## 4.2   COMPARING WITH OTHER MODELS

We benchmark OCTOCODER and OCTOGEEX with state-of-the-art Code LLMs on HUMANEVAL-PACK in Table 2. For all models, we use the prompt put forward by the model creators if applicable or else a simple intuitive prompt, see Appendix Q.

**OCTOCODER performs best among permissive models**   OCTOCODER has the highest average score across all three evaluation scenarios among all permissive models. With just 6 billion parameters, OCTOGEEX is the smallest model benchmarked, but still outperforms *all* prior permissive Code LLMs. GPT-4 (OpenAI, 2023) performs best among all models benchmarked with a significant margin. However, GPT-4 is closed-source and likely much larger than all other models evaluated.

**Instruction tuning generalizes to unseen programming languages**   Trained primarily on natural language, not code, BLOOMZ (Muennighoff et al., 2022b) performs worse than other models despite having 176 billion parameters. Go and Rust are not contained in BLOOMZ's instruction data, yet it performs much better than the random baseline of 0.0 for these two languages across most tasks. This confirms our hypothesis that models are capable of generalizing instructions to programming languages only seen at pretraining, similar to crosslingual generalization for natural languages (Muennighoff et al., 2022b). To improve programming language generalization further, we tune OCTOCODER and OCTOGEEX on many languages from COMMITPACKFT, and this generalization improvement is reflected in the performance on HUMANEVALPACK's new languages.

**Pretraining weight correlates with programming language performance after instruction tuning**
Prior work has shown that the performance on natural languages after instruction tuning is correlated with the weight of these languages during pretraining (Muennighoff et al., 2022b). The more weight during pretraining, the better the performance after instruction tuning. We find the same to be

| Model (↓) | Python | JavaScript | Java | Go | C++ | Rust | Avg. |
|---|---|---|---|---|---|---|---|
| | | | **HUMANEVALFIX** | | | | |
| | | | Non-permissive models | | | | |
| InstructCodeT5+[†] | 2.7 | 1.2 | 4.3 | 2.1 | 0.2 | 0.5 | 1.8 |
| WizardCoder[†] | 31.8 | 29.5 | 30.7 | 30.4 | 18.7 | 13.0 | 25.7 |
| GPT-4 | 47.0 | 48.2 | 50.0 | 50.6 | 47.6 | 43.3 | 47.8 |
| | | | Permissive models | | | | |
| BLOOMZ | 16.6 | 15.5 | 15.2 | 16.4 | 6.7 | 5.7 | 12.5 |
| StarChat-$\beta$ | 18.1 | 18.1 | 24.1 | 18.1 | 8.2 | 3.6 | 11.2 |
| CodeGeeX2* | 15.9 | 14.7 | 18.0 | 13.6 | 4.3 | 6.1 | 12.1 |
| StarCoder | 8.7 | 15.7 | 13.3 | 20.1 | 15.6 | 6.7 | 13.4 |
| OCTOGEEX* | 28.1 | 27.7 | 30.4 | 27.6 | 22.9 | 9.6 | 24.4 |
| OCTOCODER | **30.4** | **28.4** | **30.6** | **30.2** | **26.1** | **16.5** | **27.0** |
| | | | **HUMANEVALEXPLAIN** | | | | |
| | | | Non-permissive models | | | | |
| InstructCodeT5+[†] | 20.8 | 0.0 | 0.0 | 0.0 | 0.1 | 0.0 | 3.5 |
| WizardCoder[†] | 32.5 | 33.0 | 27.4 | 26.7 | 28.2 | 16.9 | 27.5 |
| GPT-4 | 64.6 | 57.3 | 51.2 | 58.5 | 38.4 | 42.7 | 52.1 |
| | | | Permissive models | | | | |
| BLOOMZ | 14.7 | 8.8 | 12.1 | 8.5 | 0.6 | 0.0 | 7.5 |
| StarChat-$\beta$ | 25.4 | 21.5 | 24.5 | 18.4 | 17.6 | 13.2 | 20.1 |
| CodeGeeX2* | 0.0 | 0.0 | 0.0 | 0.0 | 0.0 | 0.0 | 0.0 |
| StarCoder | 0.0 | 0.0 | 0.0 | 0.0 | 0.0 | 0.0 | 0.0 |
| OCTOGEEX* | 30.4 | 24.0 | 24.7 | **21.7** | 21.0 | **15.9** | 22.9 |
| OCTOCODER | **35.1** | **24.5** | **27.3** | 21.1 | **24.1** | 14.8 | **24.5** |
| | | | **HUMANEVALSYNTHESIZE** | | | | |
| | | | Non-permissive models | | | | |
| InstructCodeT5+[†] | 37.0 | 18.9 | 17.4 | 9.5 | 19.8 | 0.3 | 17.1 |
| WizardCoder[†] | 57.3 | 49.5 | 36.1 | 36.4 | 40.9 | 20.2 | 40.1 |
| GPT-4 | 86.6 | 82.9 | 81.7 | 72.6 | 78.7 | 67.1 | 78.3 |
| | | | Permissive models | | | | |
| BLOOMZ | 15.6 | 14.8 | 18.4 | 8.4 | 6.5 | 5.5 | 11.5 |
| StarChat-$\beta$ | 33.5 | 31.4 | 26.7 | 25.5 | 26.6 | 14.0 | 26.3 |
| CodeGeeX2* | 35.9 | 32.2 | 30.8 | 22.5 | 29.3 | 18.1 | 28.1 |
| StarCoder | 33.6 | 30.8 | 30.2 | 17.6 | 31.6 | 21.8 | 27.6 |
| OCTOGEEX* | 44.7 | 33.8 | 36.9 | 21.9 | 32.3 | 15.7 | 30.9 |
| OCTOCODER | **46.2** | **39.2** | **38.2** | **30.4** | **35.6** | **23.4** | **35.5** |

Table 2: **Zero-shot pass@1 (%) performance across HUMANEVALPACK.** InstructCodeT5+, WizardCoder, StarChat-$\beta$, StarCoder and OCTOCODER have 16B parameters. CodeGeeX2 and OCTOGEEX have 6B parameters. BLOOMZ has 176B parameters. In this work, we call models "permissive" if weights are freely accessible and usable for commercial purposes. *: Commercial license available after submitting a form. †: Trained on data that may not be used "to develop models that compete with OpenAI" thus we classify them as non-permissive in this work (see §1).

the case for programming languages. Python, Java, and JavaScript collectively make up around 30% of the pretraining data of StarCoder (Li et al., 2023b). After instruction tuning StarCoder to produce OCTOCODER, we see the best performance among these three languages, especially for HUMANEVALSYNTHESIZE. OCTOCODER performs weakest on Rust, which is the lowest resource language of StarCoder among the languages we benchmark (1.2% of pretraining data).

**Models struggle with small targeted changes**   HUMANEVALFIX is the most challenging task for most models. They commonly regenerate the buggy function without making any change (e.g. WizardCoder in Figure 34) or they introduce new bugs (e.g. GPT-4 in Figure 33). We analyze model performance by bug type in Appendix M and find bugs that require removing excess code are the most challenging. OCTOCODER performs comparatively well across all languages. Instruction tuning on COMMITPACKFT has likely taught OCTOCODER to make small, targeted changes to fix bugs.

**Models struggle switching between code and text**   Some models fail at HUMANEVALEXPLAIN, as they do not generate natural language explanations. We manually inspect explanations for the first ten samples of the Python split and disqualify a model if none of them are explanations. This is the case for StarCoder and CodeGeeX2, which generate code instead of natural language explanations. BLOOMZ and InstructCodeT5+ also occasionally generate code. Other models exclusively generate natural language explanations, not containing any code for inspected samples.

**Models struggle adhering to a specified output length**   HUMANEVALEXPLAIN instructs models to fit their explanation within a given character limit (§3). Current models appear to have no understanding of how many characters they are generating. They commonly write very short and thus underspecified explanations (e.g. BLOOMZ in Figure 35) or excessively long explanations that end up being cut off (e.g. StarChat-$\beta$ in Figure 38). Future work could investigate how to enable models to be aware of their generated output length to improve HUMANEVALEXPLAIN performance.

**HumanEval code synthesis is close to saturation**   Pure code synthesis on HUMANEVALSYN-THESIZE is the easiest task for all models. With a pass rate of 86.6% for a single solution, GPT-4 is close to fully saturating the Python subset. GPT-4 was originally found to score 67% on Python HumanEval (OpenAI, 2023) and 81% in later work (Bubeck et al., 2023). Our score for GPT-4 is significantly higher, possibly due to improvements made to the API by OpenAI, contamination of HumanEval in GPT-4 training, or slightly different prompting and evaluation. An example of our prompt is depicted in Figure 3 (right). We perform very careful evaluation to ensure every generation is correctly processed. We reproduce the HumanEval score of WizardCoder (Luo et al., 2023; Xu et al., 2023a) and find it to also perform well across other languages. For BLOOMZ and InstructCodeT5+ our evaluation leads to a higher Python score than they reported, likely because of our more careful processing of generations. OCTOCODER has the highest performance for every language among permissively licensed models. With a pass@1 of 46.2% on the original Python split, OCTOCODER improves by a relative 38% over its base model, StarCoder.

## 5   RELATED WORK

### 5.1   CODE MODELS

There has been extensive work on code models tailored to a specific coding task, such as code summarization (Iyer et al., 2016; Ahmad et al., 2020; Zhang et al., 2022a; Shi et al., 2022) or code editing (Drain et al., 2021; Zhang et al., 2022c; He et al., 2022; Zhang et al., 2022b; Wei et al., 2023; Prenner & Robbes, 2023; Fakhoury et al., 2023; Skreta et al., 2023) (also see work on edit models more generally (Reid & Neubig, 2022; Schick et al., 2022; Dwivedi-Yu et al., 2022; Raheja et al., 2023)). These works use task-specific heuristics that limit the applicability of their methods to other tasks. In contrast, we aim to build models applicable to all kinds of tasks related to code and beyond.

Through large-scale pretraining more generally applicable code models have been developed (Nijkamp et al., 2022; 2023; Xu et al., 2022a; Christopoulou et al., 2022; Gunasekar et al., 2023; Li et al., 2023b; Bui et al., 2023; Scao et al., 2022a;b). However, these models only continue code making them hard to use for tasks such as explaining code with natural language (HUMANEVALEXPLAIN). Teaching them to follow human instructions is critical to make them applicable to diverse tasks.

## 5.2    INSTRUCTION MODELS

Training models to follow instructions has led to new capabilities in text (Ouyang et al., 2022; Wang et al., 2022b; Chung et al., 2022) and visual modalities (Xu et al., 2023b; OpenAI, 2023). Prior work has shown its benefits for traditional language tasks (Wei et al., 2022; Longpre et al., 2023a; Iyer et al., 2022), multilingual tasks (Muennighoff et al., 2022b; 2024; Yong et al., 2022; Üstün et al., 2024), and dialog (Köpf et al., 2023; Bai et al., 2022; Ganguli et al., 2022). For coding applications, PanGu-Coder2 (Shen et al., 2023), WizardCoder (Luo et al., 2023) and InstructCodeT5+ (Wang et al., 2023c) are recent models trained with coding instructions. However, they all use the CodeAlpaca dataset (Chaudhary, 2023), which is synthetically generated from OpenAI models. Using data from powerful closed-source models provides a strong advantage, but limits the model use and has other limitations highlighted in §1. CoEditor (Wei et al., 2023) proposes an "auto-editing" task, trained on 1650 python commit history repositories. Our work expands this to more general coding tasks via instructions, more languages, and orders of magnitude more commit data.

## 5.3    CODE BENCHMARKS

Many code synthesis benchmarks have been proposed (Wang et al., 2022d;c; Yu et al., 2023; Lai et al., 2023; Du et al., 2023). HumanEval (Chen et al., 2021; Liu et al., 2023b) has emerged as the standard for this task. Prior work has extended HumanEval to new programming languages via automatic translation mechanisms (Athiwaratkun et al., 2022; Cassano et al., 2023; Orlanski et al., 2023). These approaches are error-prone and only translate tests, not the actual solutions, which are needed for tasks like code explanation. Thus, we rely only on humans to create all parts of HUMANEVALPACK including test cases, correct solutions, buggy solutions, and other metadata across 6 languages.

Code repair is commonly evaluated on Quixbugs (Lin et al., 2017; Prenner & Robbes, 2021; Ye et al., 2021; Xia & Zhang, 2023; Jiang et al., 2023; Sobania et al., 2023) or Python bugs (He et al., 2022; Bradley et al., 2023). The latter does not support code execution, which limits its utility. While Quixbugs supports execution with unit tests, it only contains 40 samples in Python and Java. Further, the problems in Quixbugs are generic functions, such as bucket sort. This makes them easy to solve and hard to decontaminate training data for. Our benchmark, HUMANEVALFIX, contains 164 buggy functions for six languages with solutions and unit tests. Further, our coding problems, derived from HumanEval, are very specific, such as keeping track of a bank account balance (see Figure 14).

Prior work on evaluating code explanations (Lu et al., 2021; Cui et al., 2022) has relied on metrics such as METEOR (Banerjee & Lavie, 2005) or BLEU (Papineni et al., 2002). By chaining code explanation with code synthesis, we can evaluate this task using the execution-based pass@$k$ metric overcoming the major limitations of BLEU and other heuristics-based metrics (Reiter, 2018).

Large-scale benchmarking has proven useful in many areas of natural language processing (Wang et al., 2019; Kiela et al., 2021; Srivastava et al., 2022; Muennighoff et al., 2022a). By producing 18 scores (6 languages across 3 tasks) for 9 models, we take a step towards large-scale benchmarking of code models. However, we lack many models capable of generating code (Black et al., 2021; Fried et al., 2022; Black et al., 2022; Wang & Komatsuzaki, 2021; Biderman et al., 2023b). Future work may consider more models or extending HUMANEVALPACK to new languages or tasks, such as code efficiency (Madaan et al., 2023a; Yetistiren et al., 2022) or code classification (Khan et al., 2023).

## 6    CONCLUSION

This work studies training and evaluation of Code LLMs that follow instructions. We introduce COMMITPACK, a 4TB dataset of Git commits covering 350 programming languages. We filter this large-scale dataset to create COMMITPACKFT, 2GB of high-quality code with commit messages that assimilate instructions. To enable a comprehensive evaluation of instruction code models, we construct HUMANEVALPACK, a human-written benchmark covering 3 different tasks for 6 programming languages. We ablate several instruction datasets and find that COMMITPACKFT combined with natural language data leads to the best performance. While our models, OCTOCODER and OCTOGEEX, are the best permissively licensed Code LLMs available, they are outperformed by closed-source models such as GPT-4. In addition to improving the instruction tuning paradigm, future work should consider training more capable base models.

## ACKNOWLEDGEMENTS

We thank Hugging Face for providing compute instances. We are extremely grateful to Rodrigo Garcia for the Rust translations, Dimitry Ageev and Calum Bird for help with GPT-4 evaluation, Loubna Ben Allal for help on evaluation, Arjun Guha for insightful discussions on chaining evaluation tasks to avoid evaluating with BLEU, Lewis Tunstall for help on the OASST data, Victor Sanh and Nadav Timor for discussions, Jiaxi Yang for logo editing and domain classification prompting design, Ghosal et al. (2023); Zeng et al. (2023) for design inspiration, Harm de Vries for feedback and all members of BigCode for general support. Finally, we thank every programmer who takes the time to write informative commit messages.

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
