# APPENDIX

# Contents

# A CONTRIBUTIONS

Niklas Muennighoff created COMMITPACK and HUMANEVALPACK, wrote most of the paper and led the project. Qian Liu devised many quality filters, ran SantaCoder ablations, investigated early training decisions and helped edit the paper. Armel Zebaze created the Self-Instruct data and ran numerous ablations. Niklas Muennighoff, Armel Zebaze and Qinkai Zheng created and evaluated OCTOCODER and OCTOGEEX. Binyuan Hui pretrained SantaCoder, made major contributions to the presentation and helped edit the paper. Terry Yue Zhuo ran GPT-4 evaluations and helped edit the paper. Xiangru Tang provided help on several experiments for evaluation and helped edit the paper. Leandro von Werra provided early guidance, suggested many quality filters and added the commit data to StarCoder pretraining. Niklas Muennighoff, Qian Liu, Binyuan Hui, Swayam Singh and Shayne Longpre conducted the data analysis. Shayne Longpre advised the project and made large contributions to the paper.

# B ARTIFACTS

| Model | Public Link |
|---|---|
| *Other models* | |
| Diff Codegen 2B (Bradley et al., 2023) | https://hf.co/CarperAI/diff-codegen-2b-v2 |
| InstructCodeT5+ (Wang et al., 2023c) | https://hf.co/Salesforce/instructcodet5p-16b |
| BLOOMZ (Muennighoff et al., 2022b) | https://hf.co/bigscience/bloomz |
| StarChat-$\beta$ (Tunstall et al., 2023) | https://hf.co/HuggingFaceH4/starchat-beta |
| CodeGeeX2 (Zheng et al., 2023) | https://github.com/THUDM/CodeGeeX2 |
| SantaCoder (Allal et al., 2023) | https://hf.co/bigcode/santacoder |
| StarCoder (Li et al., 2023b) | https://hf.co/bigcode/starcoder |
| WizardCoder (Luo et al., 2023) | https://hf.co/WizardLM/WizardCoder-15B-V1.0 |
| GPT-4 (OpenAI, 2023) | https://openai.com/gpt-4 |
| *Data Ablations (Appendix I) - Data* | |
| Filtered xP3x code | https://hf.co/datasets/bigcode/xp3x-octopack |
| StarCoder Self-Instruct | https://hf.co/datasets/codeparrot/self-instruct-starcoder |
| Filtered OASST | https://hf.co/datasets/bigcode/oasst-octopack |
| Manual selection (Appendix I) | https://hf.co/datasets/bigcode/co-manual |
| *Data Ablations (Appendix I) - Models* | |
| Self-Instruct (SI) | https://hf.co/bigcode/starcoder-s |
| OASST (O) | https://hf.co/bigcode/starcoder-o |
| SI + O | https://hf.co/bigcode/starcoder-so |
| xP3x + O | https://hf.co/bigcode/starcoder-xo |
| COMMITPACKFT + O (Formatting) | https://hf.co/bigcode/starcoder-co-format |
| COMMITPACKFT + O (Target loss) | https://hf.co/bigcode/starcoder-co-target |
| COMMITPACKFT + O (Manual) | https://hf.co/bigcode/starcoder-co-manual |
| COMMITPACKFT + xP3x + O | https://hf.co/bigcode/starcoder-cxo |
| COMMITPACKFT + xP3x + SI + O | https://hf.co/bigcode/starcoder-cxso |
| *SantaCoder ablations (Appendix G, Appendix J)* | |
| Commit format Pretraining | https://hf.co/bigcode/santacoderpack |
| Commit format Finetuning | https://hf.co/bigcode/santacoder-cf |
| Line diff format Finetuning | https://hf.co/bigcode/santacoder-ldf |
| *Other datasets* | |
| COMMITPACK Metadata | https://hf.co/datasets/bigcode/commitpackmeta |
| *Main artifacts* | |
| COMMITPACK | https://hf.co/datasets/bigcode/commitpack |
| COMMITPACKFT | https://hf.co/datasets/bigcode/commitpackft |
| HUMANEVALPACK | https://hf.co/datasets/bigcode/humanevalpack |
| OCTOGEEX | https://hf.co/bigcode/octogeex |
| OCTOCODER | https://hf.co/bigcode/octocoder |

Table 3: **Used and produced artifacts.**

# C  COMMITPACK AND COMMITPACKFT LANGUAGES

| Language (↓) | COMMITPACK | | | COMMITPACKFT | | |
|---|---|---|---|---|---|---|
| | MB | Samples | % (MB) | MB | Samples | % (MB) |
| Total | 3709175.78 | 57700105 | 100.0 | 1545.02 | 702062 | 100.0 |
| json | 583293.82 | 3495038 | 15.73 | 86.74 | 39777 | 5.61 |
| xml | 279208.68 | 1923159 | 7.53 | 23.68 | 9337 | 1.53 |
| text | 270662.6 | 1389525 | 7.3 | 66.66 | 46588 | 4.31 |
| javascript | 262824.84 | 5401937 | 7.09 | 125.01 | 52989 | 8.09 |
| objective-c++ | 239009.3 | 32227 | 6.44 | 0.38 | 86 | 0.02 |
| python | 234311.56 | 6189601 | 6.32 | 132.68 | 56025 | 8.59 |
| c | 200876.8 | 2779478 | 5.42 | 21.08 | 8506 | 1.36 |
| c++ | 186585.26 | 2402294 | 5.03 | 14.14 | 4992 | 0.92 |
| markdown | 171849.95 | 7645354 | 4.63 | 131.15 | 62518 | 8.49 |
| java | 127103.45 | 3744377 | 3.43 | 56.28 | 20635 | 3.64 |
| html | 105305.28 | 2366841 | 2.84 | 48.42 | 20214 | 3.13 |
| yaml | 100466.64 | 2592787 | 2.71 | 190.88 | 114320 | 12.35 |
| go | 86444.62 | 1183612 | 2.33 | 12.13 | 5004 | 0.79 |
| csv | 82946.19 | 79268 | 2.24 | 0.53 | 375 | 0.03 |
| php | 74961.64 | 2555419 | 2.02 | 60.22 | 24791 | 3.9 |
| jupyter-notebook | 66854.08 | 94000 | 1.8 | 0.1 | 48 | 0.01 |
| gettext-catalog | 62296.88 | 168327 | 1.68 | 0.13 | 72 | 0.01 |
| sql | 56802.76 | 132772 | 1.53 | 3.74 | 2069 | 0.24 |
| unity3d-asset | 39535.01 | 17867 | 1.07 | 0.16 | 101 | 0.01 |
| typescript | 39254.8 | 572136 | 1.06 | 14.28 | 5868 | 0.92 |
| owl | 36435.46 | 7458 | 0.98 | 0 | 0 | 0.0 |
| ruby | 35830.74 | 2928702 | 0.97 | 195.29 | 69413 | 12.64 |
| c# | 33669.65 | 923157 | 0.91 | 26.84 | 9346 | 1.74 |
| nix | 33547.92 | 221281 | 0.9 | 3.84 | 1593 | 0.25 |
| shell | 25109.95 | 1017977 | 0.68 | 66.86 | 31217 | 4.33 |
| perl | 21148.93 | 374266 | 0.57 | 4.99 | 2288 | 0.32 |
| tex | 17471.11 | 89283 | 0.47 | 0.56 | 307 | 0.04 |
| css | 16306.63 | 548818 | 0.44 | 9.36 | 5049 | 0.61 |
| restructuredtext | 15613.89 | 494037 | 0.42 | 15.73 | 6560 | 1.02 |
| rust | 15011.3 | 296214 | 0.4 | 7.24 | 2996 | 0.47 |
| groff | 12020.19 | 32923 | 0.32 | 0.4 | 192 | 0.03 |
| ini | 8375.16 | 297100 | 0.23 | 21.04 | 11360 | 1.36 |
| scala | 8325.96 | 316064 | 0.22 | 11.18 | 5040 | 0.72 |
| coffeescript | 6795.14 | 292446 | 0.18 | 16.96 | 5513 | 1.1 |
| haskell | 6306.12 | 217325 | 0.17 | 3.31 | 1389 | 0.21 |
| swift | 5902.72 | 319289 | 0.16 | 16.27 | 4849 | 1.05 |
| lua | 5763.12 | 139091 | 0.16 | 1.85 | 920 | 0.12 |
| svg | 5645.44 | 27095 | 0.15 | 0.25 | 169 | 0.02 |
| gas | 5585.38 | 15121 | 0.15 | 0.34 | 193 | 0.02 |
| ocaml | 5355.4 | 81360 | 0.14 | 0.7 | 333 | 0.05 |
| erlang | 5043.32 | 93685 | 0.14 | 1.19 | 480 | 0.08 |
| makefile | 4238.51 | 343379 | 0.11 | 2.53 | 960 | 0.16 |
| asciidoc | 4138.59 | 96671 | 0.11 | 1.86 | 523 | 0.12 |
| emacs-lisp | 3988.65 | 83228 | 0.11 | 1.97 | 1015 | 0.13 |
| scss | 3944.94 | 288190 | 0.11 | 13.21 | 6829 | 0.86 |
| clojure | 3523.41 | 158674 | 0.09 | 5.07 | 2403 | 0.33 |
| org | 3126.22 | 30198 | 0.08 | 0.27 | 136 | 0.02 |
| common-lisp | 2954.9 | 74628 | 0.08 | 1.45 | 778 | 0.09 |
| diff | 2586.05 | 21021 | 0.07 | 1.48 | 680 | 0.1 |
| groovy | 2569.14 | 110057 | 0.07 | 4.17 | 1486 | 0.27 |
| html+erb | 2450.68 | 225379 | 0.07 | 23.1 | 10910 | 1.5 |
| nesc | 2439.56 | 473 | 0.07 | 0.02 | 7 | 0.0 |

| | | | | | | |
|---|---|---|---|---|---|---|
| dart | 2395.8 | 56873 | 0.06 | 1.96 | 765 | 0.13 |
| powershell | 2289.28 | 55381 | 0.06 | 2.06 | 991 | 0.13 |
| f# | 2289.24 | 66840 | 0.06 | 0.66 | 254 | 0.04 |
| dm | 2223.14 | 55584 | 0.06 | 0.15 | 16 | 0.01 |
| kotlin | 2219.25 | 124266 | 0.06 | 5.37 | 2214 | 0.35 |
| pascal | 2194.68 | 42511 | 0.06 | 0.05 | 25 | 0.0 |
| jsx | 2124.74 | 139148 | 0.06 | 5.5 | 2199 | 0.36 |
| viml | 1948.21 | 74062 | 0.05 | 1.96 | 1063 | 0.13 |
| actionscript | 1844.15 | 28819 | 0.05 | 0.12 | 49 | 0.01 |
| cython | 1736.59 | 25927 | 0.05 | 0.31 | 123 | 0.02 |
| turtle | 1698.95 | 3882 | 0.05 | 0.05 | 21 | 0.0 |
| less | 1616.56 | 88634 | 0.04 | 3.72 | 1360 | 0.24 |
| mathematica | 1475.04 | 925 | 0.04 | 0.01 | 1 | 0.0 |
| xslt | 1441.46 | 27956 | 0.04 | 0.26 | 99 | 0.02 |
| scheme | 1249.24 | 30546 | 0.03 | 0.42 | 213 | 0.03 |
| perl6 | 1223.16 | 12167 | 0.03 | 0.27 | 122 | 0.02 |
| edn | 1186.94 | 2289 | 0.03 | 0.09 | 48 | 0.01 |
| fortran | 1178.55 | 13463 | 0.03 | 0.14 | 70 | 0.01 |
| java-server-pages | 1173.07 | 53574 | 0.03 | 0.45 | 173 | 0.03 |
| standard-ml | 1133.48 | 20097 | 0.03 | 0.15 | 72 | 0.01 |
| cmake | 1132.07 | 58446 | 0.03 | 2.27 | 981 | 0.15 |
| json5 | 1108.2 | 1827 | 0.03 | 0.08 | 33 | 0.01 |
| vala | 1104.51 | 14822 | 0.03 | 0.12 | 50 | 0.01 |
| vue | 1093.8 | 68967 | 0.03 | 1.38 | 587 | 0.09 |
| freemarker | 1032.33 | 36216 | 0.03 | 1.03 | 510 | 0.07 |
| graphql | 1004.84 | 2009 | 0.03 | 0.03 | 17 | 0.0 |
| twig | 958.96 | 39588 | 0.03 | 3.96 | 1610 | 0.26 |
| tcl | 869.83 | 16407 | 0.02 | 0.29 | 103 | 0.02 |
| pod | 859.02 | 14922 | 0.02 | 0.15 | 54 | 0.01 |
| dockerfile | 849.73 | 259379 | 0.02 | 0.1 | 39 | 0.01 |
| yacc | 845.7 | 8230 | 0.02 | 0.01 | 3 | 0.0 |
| postscript | 800.73 | 903 | 0.02 | 0.02 | 9 | 0.0 |
| racket | 796.64 | 16615 | 0.02 | 0.2 | 117 | 0.01 |
| eagle | 785.68 | 2237 | 0.02 | 0.01 | 4 | 0.0 |
| haxe | 772.9 | 28447 | 0.02 | 0.34 | 174 | 0.02 |
| julia | 752.07 | 22695 | 0.02 | 0.31 | 180 | 0.02 |
| handlebars | 740.82 | 49842 | 0.02 | 3.29 | 1429 | 0.21 |
| smarty | 720.94 | 41065 | 0.02 | 1.59 | 737 | 0.1 |
| visual-basic | 681.52 | 10511 | 0.02 | 0.15 | 48 | 0.01 |
| literate-haskell | 673.74 | 10729 | 0.02 | 0.02 | 7 | 0.0 |
| smalltalk | 665.89 | 11741 | 0.02 | 0.46 | 284 | 0.03 |
| isabelle | 655.82 | 8359 | 0.02 | 0.01 | 2 | 0.0 |
| nimrod | 652.86 | 12023 | 0.02 | 0.24 | 67 | 0.02 |
| zig | 621.38 | 4290 | 0.02 | 0.01 | 4 | 0.0 |
| m4 | 603.58 | 12465 | 0.02 | 0.26 | 101 | 0.02 |
| max | 603.56 | 2259 | 0.02 | 0 | 0 | 0.0 |
| elixir | 558.12 | 35473 | 0.02 | 2.35 | 1150 | 0.15 |
| mako | 543.01 | 8943 | 0.01 | 0.76 | 170 | 0.05 |
| arduino | 534.18 | 32350 | 0.01 | 0.46 | 225 | 0.03 |
| jade | 531.4 | 46993 | 0.01 | 2.35 | 1119 | 0.15 |
| haml | 502.01 | 74792 | 0.01 | 10.74 | 4415 | 0.7 |
| elm | 481.97 | 18542 | 0.01 | 0.62 | 265 | 0.04 |
| purebasic | 474.28 | 36 | 0.01 | 0.02 | 5 | 0.0 |
| coldfusion | 470.78 | 9263 | 0.01 | 0.02 | 9 | 0.0 |
| lean | 470.03 | 7507 | 0.01 | 0.02 | 3 | 0.0 |
| r | 454.32 | 12858 | 0.01 | 0.23 | 121 | 0.01 |
| cuda | 437.67 | 11450 | 0.01 | 0.07 | 25 | 0.0 |
| textile | 425.12 | 18491 | 0.01 | 0.18 | 61 | 0.01 |
| robotframework | 421.61 | 9211 | 0.01 | 0.21 | 85 | 0.01 |

| | | | | | | |
|---|---|---|---|---|---|---|
| abap | 409.62 | 1955 | 0.01 | 0.01 | 1 | 0.0 |
| rdoc | 397.03 | 38760 | 0.01 | 0.55 | 270 | 0.04 |
| llvm | 382.2 | 10727 | 0.01 | 1.6 | 780 | 0.1 |
| ada | 380.7 | 13258 | 0.01 | 0.73 | 265 | 0.05 |
| batchfile | 372.16 | 43674 | 0.01 | 2.98 | 1466 | 0.19 |
| qml | 361.45 | 19360 | 0.01 | 0.94 | 368 | 0.06 |
| jasmin | 359.82 | 4782 | 0.01 | 0.05 | 9 | 0.0 |
| assembly | 343.62 | 8126 | 0.01 | 0.17 | 105 | 0.01 |
| g-code | 334.96 | 3690 | 0.01 | 0.04 | 7 | 0.0 |
| cucumber | 331.38 | 26677 | 0.01 | 2.59 | 976 | 0.17 |
| html+php | 323.35 | 18381 | 0.01 | 0.33 | 150 | 0.02 |
| kicad | 321.94 | 759 | 0.01 | 0 | 0 | 0.0 |
| api-blueprint | 317.85 | 4765 | 0.01 | 0.06 | 23 | 0.0 |
| eiffel | 311.48 | 373 | 0.01 | 0.01 | 2 | 0.0 |
| toml | 292.68 | 63517 | 0.01 | 5.58 | 3424 | 0.36 |
| modelica | 284.62 | 2611 | 0.01 | 0.04 | 15 | 0.0 |
| bitbake | 277.58 | 43239 | 0.01 | 4.46 | 1308 | 0.29 |
| lex | 275.96 | 705 | 0.01 | 0 | 0 | 0.0 |
| stylus | 273.06 | 21967 | 0.01 | 0.95 | 480 | 0.06 |
| protocol-buffer | 254.12 | 9202 | 0.01 | 0.52 | 181 | 0.03 |
| unknown | 252.23 | 30570 | 0.01 | 3.05 | 1597 | 0.2 |
| nit | 244.54 | 4951 | 0.01 | 0.02 | 3 | 0.0 |
| factor | 241.19 | 15378 | 0.01 | 0.36 | 113 | 0.02 |
| xs | 239.04 | 3215 | 0.01 | 0.02 | 7 | 0.0 |
| sass | 230.65 | 23144 | 0.01 | 1.36 | 705 | 0.09 |
| pir | 230.2 | 6231 | 0.01 | 0.08 | 23 | 0.01 |
| html+django | 217.04 | 10535 | 0.01 | 0.85 | 399 | 0.06 |
| mediawiki | 214.32 | 10188 | 0.01 | 0.08 | 33 | 0.01 |
| logos | 212.3 | 1733 | 0.01 | 0.04 | 19 | 0.0 |
| genshi | 209.3 | 956 | 0.01 | 0.02 | 3 | 0.0 |
| coldfusion-cfc | 208.16 | 4410 | 0.01 | 0.05 | 20 | 0.0 |
| xtend | 179.54 | 7775 | 0.0 | 0.13 | 55 | 0.01 |
| sqf | 168.66 | 7778 | 0.0 | 0.09 | 45 | 0.01 |
| vhdl | 155.95 | 2185 | 0.0 | 0.02 | 5 | 0.0 |
| antlr | 143.55 | 3651 | 0.0 | 0.03 | 15 | 0.0 |
| systemverilog | 140.19 | 3944 | 0.0 | 0.08 | 35 | 0.01 |
| hcl | 136.75 | 13379 | 0.0 | 0.91 | 421 | 0.06 |
| asp | 136.1 | 4286 | 0.0 | 0.09 | 22 | 0.01 |
| nsis | 129.12 | 4048 | 0.0 | 0.06 | 15 | 0.0 |
| inform-7 | 120.19 | 184 | 0.0 | 0.01 | 2 | 0.0 |
| slim | 119.04 | 18726 | 0.0 | 2.06 | 1052 | 0.13 |
| groovy-server-pages | 117.37 | 6695 | 0.0 | 0.07 | 25 | 0.0 |
| ceylon | 116.14 | 7256 | 0.0 | 0.1 | 49 | 0.01 |
| fish | 111.28 | 15351 | 0.0 | 1.33 | 813 | 0.09 |
| processing | 108.58 | 5912 | 0.0 | 0.07 | 35 | 0.0 |
| component-pascal | 105.5 | 43 | 0.0 | 0 | 0 | 0.0 |
| lasso | 104.17 | 67 | 0.0 | 0 | 0 | 0.0 |
| glsl | 99.49 | 9478 | 0.0 | 0.34 | 164 | 0.02 |
| saltstack | 98.2 | 12314 | 0.0 | 1.41 | 617 | 0.09 |
| xbase | 94.42 | 1670 | 0.0 | 0.01 | 3 | 0.0 |
| autohotkey | 94.22 | 1452 | 0.0 | 0.02 | 15 | 0.0 |
| liquid | 93.79 | 2651 | 0.0 | 0.09 | 30 | 0.01 |
| purescript | 92.41 | 5024 | 0.0 | 0.17 | 80 | 0.01 |
| agda | 92.06 | 4956 | 0.0 | 0.02 | 10 | 0.0 |
| inno-setup | 91.36 | 3014 | 0.0 | 0.06 | 16 | 0.0 |
| oz | 90.48 | 1551 | 0.0 | 0.03 | 8 | 0.0 |
| chapel | 89.62 | 26447 | 0.0 | 0.04 | 20 | 0.0 |
| arc | 87.21 | 758 | 0.0 | 0.01 | 2 | 0.0 |
| opencl | 86.43 | 2489 | 0.0 | 0.05 | 23 | 0.0 |

| | | | | | | |
|---|---|---|---|---|---|---|
| graphviz-dot | 85.8 | 1525 | 0.0 | 0.07 | 35 | 0.0 |
| pawn | 85.42 | 580 | 0.0 | 0.01 | 3 | 0.0 |
| jsoniq | 75.15 | 1343 | 0.0 | 0.01 | 6 | 0.0 |
| bluespec | 72.38 | 2500 | 0.0 | 0.01 | 2 | 0.0 |
| smali | 71.38 | 174 | 0.0 | 0 | 0 | 0.0 |
| krl | 69.87 | 1879 | 0.0 | 0.02 | 4 | 0.0 |
| maple | 68.28 | 1311 | 0.0 | 0.01 | 2 | 0.0 |
| unrealscript | 67.67 | 585 | 0.0 | 0.01 | 1 | 0.0 |
| ooc | 63.19 | 3416 | 0.0 | 0.04 | 15 | 0.0 |
| pure-data | 62.62 | 603 | 0.0 | 0.01 | 1 | 0.0 |
| xquery | 61.96 | 2237 | 0.0 | 0.08 | 39 | 0.01 |
| dcl | 59.64 | 833 | 0.0 | 0.04 | 19 | 0.0 |
| moonscript | 59.21 | 1951 | 0.0 | 0.02 | 10 | 0.0 |
| awk | 57.18 | 2206 | 0.0 | 0.1 | 52 | 0.01 |
| pike | 52.87 | 1262 | 0.0 | 0.02 | 6 | 0.0 |
| livescript | 51.23 | 5194 | 0.0 | 0.13 | 63 | 0.01 |
| solidity | 50.86 | 3689 | 0.0 | 0.08 | 37 | 0.01 |
| monkey | 48.26 | 1367 | 0.0 | 0.02 | 4 | 0.0 |
| jsonld | 48.01 | 462 | 0.0 | 0.02 | 6 | 0.0 |
| zephir | 42.68 | 1265 | 0.0 | 0.02 | 4 | 0.0 |
| crystal | 41.92 | 4217 | 0.0 | 0.35 | 182 | 0.02 |
| rhtml | 41.02 | 4551 | 0.0 | 0.35 | 135 | 0.02 |
| stata | 40.68 | 1344 | 0.0 | 0.02 | 10 | 0.0 |
| idris | 39.9 | 3025 | 0.0 | 0.13 | 38 | 0.01 |
| raml | 39.39 | 948 | 0.0 | 0.03 | 9 | 0.0 |
| openscad | 37.73 | 2178 | 0.0 | 0.05 | 21 | 0.0 |
| red | 35.26 | 1108 | 0.0 | 0.01 | 1 | 0.0 |
| c2hs-haskell | 34.47 | 1021 | 0.0 | 0.01 | 2 | 0.0 |
| cycript | 33.96 | 197 | 0.0 | 0 | 0 | 0.0 |
| applescript | 33.51 | 1304 | 0.0 | 0.04 | 19 | 0.0 |
| mupad | 32.49 | 178 | 0.0 | 0.02 | 4 | 0.0 |
| literate-agda | 31.38 | 567 | 0.0 | 0.01 | 1 | 0.0 |
| boo | 31.17 | 26289 | 0.0 | 0.01 | 2 | 0.0 |
| sourcepawn | 29.53 | 717 | 0.0 | 0.01 | 3 | 0.0 |
| qmake | 29.51 | 3632 | 0.0 | 0.32 | 140 | 0.02 |
| ragel-in-ruby-host | 28.3 | 888 | 0.0 | 0.01 | 4 | 0.0 |
| io | 27.95 | 1247 | 0.0 | 0.01 | 4 | 0.0 |
| desktop | 27.65 | 5021 | 0.0 | 0.36 | 186 | 0.02 |
| propeller-spin | 26.77 | 625 | 0.0 | 0.01 | 1 | 0.0 |
| thrift | 26.75 | 1007 | 0.0 | 0.08 | 28 | 0.01 |
| volt | 25.05 | 1660 | 0.0 | 0.02 | 9 | 0.0 |
| xproc | 24.21 | 914 | 0.0 | 0.02 | 3 | 0.0 |
| igor-pro | 23.75 | 388 | 0.0 | 0.01 | 1 | 0.0 |
| lolcode | 23.74 | 24861 | 0.0 | 0 | 0 | 0.0 |
| html+eex | 21.41 | 2100 | 0.0 | 0.29 | 135 | 0.02 |
| logtalk | 20.43 | 1035 | 0.0 | 0.06 | 21 | 0.0 |
| mirah | 20.1 | 706 | 0.0 | 0.04 | 16 | 0.0 |
| gnuplot | 19.68 | 889 | 0.0 | 0.03 | 17 | 0.0 |
| literate-coffeescript | 19.02 | 1041 | 0.0 | 0.05 | 19 | 0.0 |
| jflex | 18.61 | 555 | 0.0 | 0.01 | 1 | 0.0 |
| emberscript | 18.39 | 1024 | 0.0 | 0.02 | 7 | 0.0 |
| cobol | 17.0 | 24953 | 0.0 | 0 | 0 | 0.0 |
| yang | 16.94 | 597 | 0.0 | 0.02 | 6 | 0.0 |
| rebol | 16.47 | 239 | 0.0 | 0.01 | 3 | 0.0 |
| linker-script | 16.08 | 1604 | 0.0 | 0.08 | 37 | 0.01 |
| cartocss | 15.92 | 555 | 0.0 | 0.01 | 3 | 0.0 |
| urweb | 13.07 | 304 | 0.0 | 0.02 | 6 | 0.0 |
| rmarkdown | 13.03 | 750 | 0.0 | 0 | 0 | 0.0 |
| darcs-patch | 13.01 | 80 | 0.0 | 0 | 0 | 0.0 |

| | | | | | | |
|---|---|---|---|---|---|---|
| csound | 12.85 | 229 | 0.0 | 0.01 | 4 | 0.0 |
| squirrel | 12.84 | 531 | 0.0 | 0.01 | 4 | 0.0 |
| apl | 12.56 | 586 | 0.0 | 0.02 | 7 | 0.0 |
| hlsl | 12.17 | 1529 | 0.0 | 0.03 | 11 | 0.0 |
| latte | 11.89 | 1380 | 0.0 | 0.02 | 7 | 0.0 |
| pony | 11.84 | 624 | 0.0 | 0.05 | 16 | 0.0 |
| ioke | 10.86 | 373 | 0.0 | 0.04 | 25 | 0.0 |
| hy | 10.51 | 879 | 0.0 | 0.04 | 12 | 0.0 |
| uno | 10.36 | 628 | 0.0 | 0.01 | 2 | 0.0 |
| pan | 10.34 | 637 | 0.0 | 0.05 | 23 | 0.0 |
| xojo | 10.31 | 642 | 0.0 | 0 | 0 | 0.0 |
| papyrus | 10.26 | 130 | 0.0 | 0 | 0 | 0.0 |
| stan | 10.25 | 540 | 0.0 | 0 | 0 | 0.0 |
| slash | 9.9 | 640 | 0.0 | 0.01 | 4 | 0.0 |
| supercollider | 9.8 | 318 | 0.0 | 0.01 | 2 | 0.0 |
| vcl | 9.46 | 747 | 0.0 | 0.04 | 18 | 0.0 |
| smt | 9.03 | 117 | 0.0 | 0.01 | 3 | 0.0 |
| glyph | 8.95 | 7 | 0.0 | 0 | 0 | 0.0 |
| wisp | 8.74 | 262 | 0.0 | 0.01 | 3 | 0.0 |
| renpy | 8.3 | 421 | 0.0 | 0.02 | 3 | 0.0 |
| clips | 7.73 | 450 | 0.0 | 0 | 0 | 0.0 |
| dns-zone | 7.56 | 54 | 0.0 | 0.01 | 2 | 0.0 |
| sas | 7.54 | 269 | 0.0 | 0.01 | 1 | 0.0 |
| rouge | 7.2 | 396 | 0.0 | 0.1 | 41 | 0.01 |
| ec | 7.03 | 94 | 0.0 | 0 | 0 | 0.0 |
| dylan | 6.82 | 280 | 0.0 | 0.01 | 2 | 0.0 |
| tcsh | 6.52 | 748 | 0.0 | 0.02 | 10 | 0.0 |
| aspectj | 6.33 | 451 | 0.0 | 0.02 | 8 | 0.0 |
| netlogo | 6.3 | 140 | 0.0 | 0 | 0 | 0.0 |
| gap | 6.1 | 46 | 0.0 | 0 | 0 | 0.0 |
| fancy | 5.95 | 675 | 0.0 | 0.02 | 8 | 0.0 |
| coq | 5.74 | 80 | 0.0 | 0 | 0 | 0.0 |
| click | 5.74 | 9 | 0.0 | 0 | 0 | 0.0 |
| capn-proto | 5.64 | 330 | 0.0 | 0.04 | 12 | 0.0 |
| flux | 5.57 | 47 | 0.0 | 0.01 | 3 | 0.0 |
| forth | 5.51 | 265 | 0.0 | 0.01 | 2 | 0.0 |
| ats | 5.42 | 383 | 0.0 | 0.01 | 3 | 0.0 |
| netlinx | 5.17 | 144 | 0.0 | 0.01 | 1 | 0.0 |
| clean | 5.07 | 171 | 0.0 | 0.01 | 1 | 0.0 |
| parrot-assembly | 4.66 | 227 | 0.0 | 0.01 | 2 | 0.0 |
| alloy | 4.64 | 203 | 0.0 | 0 | 0 | 0.0 |
| lfe | 4.58 | 287 | 0.0 | 0.02 | 6 | 0.0 |
| gdscript | 4.49 | 460 | 0.0 | 0.03 | 9 | 0.0 |
| augeas | 4.44 | 395 | 0.0 | 0.04 | 13 | 0.0 |
| sparql | 4.4 | 1036 | 0.0 | 0.04 | 23 | 0.0 |
| lilypond | 4.31 | 265 | 0.0 | 0.01 | 6 | 0.0 |
| scilab | 4.09 | 375 | 0.0 | 0.02 | 10 | 0.0 |
| autoit | 4.06 | 279 | 0.0 | 0 | 0 | 0.0 |
| myghty | 3.86 | 105 | 0.0 | 0 | 0 | 0.0 |
| blitzmax | 3.74 | 220 | 0.0 | 0.01 | 1 | 0.0 |
| creole | 3.42 | 337 | 0.0 | 0.01 | 2 | 0.0 |
| harbour | 3.34 | 107 | 0.0 | 0.01 | 1 | 0.0 |
| piglatin | 3.17 | 513 | 0.0 | 0.02 | 11 | 0.0 |
| opa | 3.16 | 211 | 0.0 | 0 | 0 | 0.0 |
| sage | 3.03 | 414 | 0.0 | 0.01 | 1 | 0.0 |
| ston | 2.85 | 414 | 0.0 | 0.01 | 6 | 0.0 |
| maxscript | 2.8 | 47 | 0.0 | 0 | 0 | 0.0 |
| lsl | 2.68 | 74 | 0.0 | 0.01 | 3 | 0.0 |
| gentoo-ebuild | 2.58 | 601 | 0.0 | 0.06 | 16 | 0.0 |

| | | | | | | |
|---|---|---|---|---|---|---|
| nu | 2.38 | 170 | 0.0 | 0.01 | 2 | 0.0 |
| bro | 2.34 | 333 | 0.0 | 0.01 | 3 | 0.0 |
| xc | 2.02 | 88 | 0.0 | 0 | 0 | 0.0 |
| j | 1.81 | 142 | 0.0 | 0 | 0 | 0.0 |
| metal | 1.72 | 151 | 0.0 | 0.02 | 4 | 0.0 |
| mms | 1.54 | 91 | 0.0 | 0.01 | 1 | 0.0 |
| webidl | 1.51 | 96 | 0.0 | 0.05 | 6 | 0.0 |
| tea | 1.47 | 29 | 0.0 | 0 | 0 | 0.0 |
| redcode | 1.27 | 149 | 0.0 | 0 | 0 | 0.0 |
| shen | 1.2 | 71 | 0.0 | 0 | 0 | 0.0 |
| pov-ray-sdl | 1.14 | 104 | 0.0 | 0.01 | 5 | 0.0 |
| x10 | 1.01 | 33 | 0.0 | 0 | 0 | 0.0 |
| brainfuck | 0.96 | 167 | 0.0 | 0.01 | 2 | 0.0 |
| ninja | 0.95 | 187 | 0.0 | 0.03 | 14 | 0.0 |
| golo | 0.9 | 115 | 0.0 | 0 | 0 | 0.0 |
| webassembly | 0.86 | 83 | 0.0 | 0 | 0 | 0.0 |
| self | 0.82 | 15 | 0.0 | 0 | 0 | 0.0 |
| labview | 0.81 | 61 | 0.0 | 0 | 0 | 0.0 |
| octave | 0.8 | 12 | 0.0 | 0 | 0 | 0.0 |
| pogoscript | 0.8 | 74 | 0.0 | 0 | 0 | 0.0 |
| d | 0.8 | 20 | 0.0 | 0 | 0 | 0.0 |
| http | 0.74 | 140 | 0.0 | 0.03 | 19 | 0.0 |
| ecl | 0.66 | 48 | 0.0 | 0.01 | 4 | 0.0 |
| chuck | 0.58 | 99 | 0.0 | 0 | 0 | 0.0 |
| gosu | 0.52 | 60 | 0.0 | 0 | 0 | 0.0 |
| parrot | 0.52 | 17 | 0.0 | 0 | 0 | 0.0 |
| opal | 0.47 | 69 | 0.0 | 0 | 0 | 0.0 |
| objective-j | 0.46 | 37 | 0.0 | 0 | 0 | 0.0 |
| kit | 0.41 | 48 | 0.0 | 0 | 0 | 0.0 |
| gams | 0.38 | 18 | 0.0 | 0 | 0 | 0.0 |
| prolog | 0.28 | 35 | 0.0 | 0 | 0 | 0.0 |
| clarion | 0.27 | 13 | 0.0 | 0 | 0 | 0.0 |
| mask | 0.25 | 37 | 0.0 | 0.01 | 4 | 0.0 |
| brightscript | 0.24 | 28 | 0.0 | 0 | 0 | 0.0 |
| scaml | 0.18 | 31 | 0.0 | 0.01 | 1 | 0.0 |
| matlab | 0.16 | 29 | 0.0 | 0 | 0 | 0.0 |
| idl | 0.15 | 1 | 0.0 | 0 | 0 | 0.0 |
| ags-script | 0.12 | 31 | 0.0 | 0 | 0 | 0.0 |
| lookml | 0.12 | 10 | 0.0 | 0 | 0 | 0.0 |
| apacheconf | 0.11 | 59 | 0.0 | 0.01 | 2 | 0.0 |
| oxygene | 0.1 | 9 | 0.0 | 0 | 0 | 0.0 |
| txl | 0.1 | 3 | 0.0 | 0 | 0 | 0.0 |
| gf | 0.09 | 39 | 0.0 | 0 | 0 | 0.0 |
| renderscript | 0.06 | 54 | 0.0 | 0 | 0 | 0.0 |
| mtml | 0.05 | 13 | 0.0 | 0.01 | 2 | 0.0 |
| unified-parallel-c | 0.05 | 6 | 0.0 | 0 | 0 | 0.0 |
| dogescript | 0.04 | 10 | 0.0 | 0 | 0 | 0.0 |
| gentoo-eclass | 0.04 | 6 | 0.0 | 0 | 0 | 0.0 |
| zimpl | 0.04 | 7 | 0.0 | 0 | 0 | 0.0 |
| irc-log | 0.04 | 9 | 0.0 | 0 | 0 | 0.0 |
| fantom | 0.03 | 11 | 0.0 | 0 | 0 | 0.0 |
| numpy | 0.03 | 1 | 0.0 | 0 | 0 | 0.0 |
| cirru | 0.02 | 4 | 0.0 | 0 | 0 | 0.0 |
| xpages | 0.02 | 7 | 0.0 | 0.01 | 1 | 0.0 |
| nginx | 0.02 | 6 | 0.0 | 0.01 | 2 | 0.0 |
| objdump | 0.02 | 1 | 0.0 | 0 | 0 | 0.0 |
| python-traceback | 0.02 | 10 | 0.0 | 0 | 0 | 0.0 |
| realbasic | 0.01 | 1 | 0.0 | 0 | 0 | 0.0 |
| befunge | 0.01 | 2 | 0.0 | 0 | 0 | 0.0 |

| | | | | | | |
|---|---|---|---|---|---|---|
| bison | 0.01 | 1 | 0.0 | 0 | 0 | 0.0 |
| m | 0.01 | 1 | 0.0 | 0 | 0 | 0.0 |
| omgrofl | 0.01 | 1 | 0.0 | 0 | 0 | 0.0 |

Table 4: **Programming language distribution of COMMITPACK and COMMITPACKFT.** Shortcuts: MB=Megabytes, owl=web-ontology-language, pir=parrot-internal-representation, dcl=digital-command-language, mms=module-management-system, gf=grammatical-framework

# D   DATASET CREATION

**COMMITPACK**   We use the GitHub archive available on GCP which contains metadata from GitHub commits up to 2016.[4] It contains around 3TB of GitHub activity data for more than 2.8 million GitHub repositories including more than 145 million unique commits, over 2 billion different file paths and the contents of the latest revision for 163 million files.[5] We apply the filters in Table 5 to this dataset. The resulting dataset containing only metadata is uploaded at `https://hf.co/datasets/big code/commitpackmeta`. As the activity dataset only contains commit ids without the actual code changes, we scrape the code from GitHub. We use the metadata and the GitHub API to scrape the changed file prior and after the respective commit. Some repositories referenced in the activity data are no longer accessible, thus we discard them. This results in COMMITPACK with approximately 4 terabytes uploaded at `https://hf.co/datasets/bigcode/commitpack`.

| Description | Details |
|---|---|
| License | Only keep samples licensed as MIT, Artistic-2.0, ISC, CC0-1.0, EPL-1.0, MPL-2.0, Apache-2.0, BSD-3-Clause, AGPL-3.0, LGPL-2.1, BSD-2-Clause or without license. |
| Length | Only keep code where the commit message has at least 5 and at most 10,000 characters |
| Noise | Remove code where the lowercased commit message is any of 'add files via upload', "can't you see i'm updating the time?", 'commit', 'create readme.md', 'dummy', 'first commit', 'heartbeat update', 'initial commit', 'mirroring from micro.blog.', 'no message', 'pi push', 'readme', 'update', 'updates', 'update _config.yaml', 'update index.html', 'update readme.md', 'update readme', 'updated readme', 'update log', 'update data.js', 'update data.json', 'update data.js', 'pi push' or starts with 'merge' |
| Single file | Remove samples that contain changes across multiple files |
| Opt-out | Remove samples from repositories owned by users that opted out of The Stack (Kocetkov et al., 2022) |

Table 5: **COMMITPACK filters.**

**COMMITPACKFT**   Prior work has shown the importance of careful data filtering to maintain quality (Yin et al., 2018; Dhole et al., 2021; Laurençon et al., 2022; Longpre et al., 2023b; Singh et al., 2024). To create a smaller version focused on commits that resemble high-quality instructions, we further filter COMMITPACK to create COMMITPACKFT using the steps outlined in Table 6. We also checked for any contamination with HumanEval (Chen et al., 2021) but did not find any solution or docstring present in COMMITPACKFT. This is likely because our commit data only goes up to 2016, which is several years prior to the release of HumanEval. Our filters reduce the dataset by a factor of around 1000 resulting in close to 2 gigabytes across 277 languages. To gain a deeper understanding of the rich content within COMMITPACKFT, we analyze commits on its Python subset (56K samples). We first collect the most prevalent commit domain by prompting GPT-4 with: `"I'd like to know the main types of commits on Github and aim to cover as comprehensively as possible."`. Subsequently, we use GPT-4 to classify each sample using the prompt in Figure 5. The task distribution is visualized in Figure 2.

---

[4]https://www.gharchive.org/
[5]https://github.blog/2016-06-29-making-open-source-data-more-available/

| Description | Details |
|---|---|
| Length | Remove samples where the before code has more than 50,000 characters |
| Length | Remove samples where the after code has 0 characters |
| Difference | Remove samples where the before and after code are the same (e.g. file name changes) |
| Difference | Remove samples that contain a hashtag (to avoid references to issues) |
| Extension | Remove samples where the filename of the code after has an atypical extension for the programming language (e.g. only keep '.py' for Python) |
| Filename | Remove samples where the filename is contained in the commit message (as we do not use the filename in finetuning) |
| Length | Only keep samples where the commit message has more than 10 and less than 1000 characters |
| Words | Only keep samples where the commit message can be split into more than 4 and less than 1000 space-separated words |
| Clean | Remove any appearances of '[skip ci]', '[ci skip]', sequences at the beginning or end that are in brackets, sequences at the beginning that end with ':' and strip whitespace at the beginning or end |
| Capitalized | Only keep samples where the message starts with an uppercase letter |
| Tokens | Only keep samples where the concatenation of the code before, a special token and the code after has at least 50 tokens and at most 768 tokens according to the StarCoder tokenizer |
| Instructions | Only keep samples where the lowercased commit message starts with any of the words in Table 7 |
| Noise | Remove samples where the lowercased commit message contains any of 'auto commit', 'update contributing', '<?xml', 'merge branch', 'merge pull request', 'signed-off-by', "fix that bug where things didn't work but now they should", "put the thingie in the thingie", "add a beter commit message", "code review", "//codereview", "work in progress", "wip", "https://", "http://", "\| leetcode", "cdpcp", " i ", "i've" , "i'm" or both "thanks to" and "for" |
| Regex | Remove samples where the lowercased commit message has a match for any of the regular expressions `(?:v)?\d+\.\d+\.\d+(?=$|\S)`, `^[a-f0-9]+(?:-[a-f0-9]+)*$`, `([a-f0-9]{40})`, `issue\s*\d+`, `bug\s*\d+` or `feature\s*\d+` |
| Downsample | With 90% probability remove samples where the commit message starts with "Bump", "Set version" or "Update version" |

Table 6: **COMMITPACKFT filters applied to COMMITPACK.** With the commit message we refer to the commit message subject only, not the body.

"abort', 'accelerate', 'access', 'accumulate', 'add', 'address', 'adjust', 'advance', 'align', 'allot', 'allow', 'amplify', 'annotate', 'append', 'apply', 'archive', 'arrange', 'attach', 'augment', 'automate', 'backup', 'boost', 'break', 'bring', 'brush up', 'build', 'bump', 'call', 'change', 'check', 'choose', 'clarify', 'clean', 'clear', 'clone', 'comment', 'complete', 'compress', 'concatenate', 'configure', 'connect', 'consolidate', 'convert', 'copy', 'correct', 'cover', 'create', 'customize', 'cut', 'deal with', 'debug', 'decipher', 'declare', 'decommission', 'decomplexify', 'decompress', 'decrease', 'decrypt', 'define', 'delete', 'deploy', 'designate', 'destroy', 'detach', 'determine', 'develop', 'diminish', 'disable', 'discard', 'disentangle', 'dismantle', 'divide', 'document', 'downgrade', 'drop', 'duplicate', 'edit', 'embed', 'emphasize', 'enable', 'encrypt', 'enforce', 'enhance', 'enlarge', 'enumerate', 'eradicate', 'escalate', 'establish', 'exclude', 'exit', 'expand', 'expedite', 'expire', 'extend', 'facilitate', 'fix', 'format', 'gather', 'generalize', 'halt', 'handle', 'hasten', 'hide', 'implement', 'improve', 'include', 'increase', 'increment', 'indent', 'index', 'inflate', 'initialize', 'insert', 'install', 'integrate', 'interpolate', 'interrupt', 'introduce', 'isolate', 'join', 'kill', 'leverage', 'load', 'magnify', 'maintain', 'make', 'manage', 'mark', 'mask', 'mend', 'merge', 'migrate', 'modify', 'monitor', 'move', 'multiply', 'normalize', 'optimize', 'orchestrate', 'order', 'package', 'paraphrase', 'paste', 'patch', 'plug ', 'prepare', 'prepend', 'print', 'provision', 'purge', 'put', 'quit', 'raise', 'read', 'reannotate', 'rearrange', 'rebase', 'reboot', 'rebuild', 'recomment', 'recompile', 'reconfigure', 'reconnect', 'rectify', 'redact', 'redefine', 'reduce', 'refactor', 'reformat', 'refresh', 'reimplement', 'reinforce', 'relocate', 'remove', 'rename', 'reorder', 'reorganize', 'repackage', 'repair', 'rephrase', 'replace', 'reposition', 'reschedule', 'reset', 'reshape', 'resolve', 'restructure', 'return', 'revert', 'revise', 'revoke', 'reword', 'rework', 'rewrite', 'rollback', 'save', 'scale', 'scrub', 'secure', 'select', 'send', 'set', 'settle', 'simplify', 'solve', 'sort', 'speed up', 'split', 'stabilize', 'standardize', 'stipulate', 'stop', 'store', 'streamline', 'strengthen', 'structure', 'substitute', 'subtract', 'support', 'swap', 'switch', 'synchronize', 'tackle', 'tag', 'terminate', 'test', 'throw', 'tidy', 'transform', 'transpose', 'trim', 'troubleshoot', 'truncate', 'tweak', 'unblock', 'uncover', 'undo', 'unify', 'uninstall', 'unplug', 'unpublish', 'unravel', 'unstage', 'unsync', 'untangle', 'unwind', 'update', 'upgrade', 'use', 'validate', 'verify', 'watch', 'watermark', 'whitelist', 'withdraw', 'work', 'write"

Table 7: **Commit message starting words allowed in COMMITPACKFT.**

---

Please categorize the following commit message, which may fall into more than one category.

### Category
Bug fixes, New features, Refactoring/code cleanup, Documentation, Testing, User interface, Dependencies, Configuration, Build system/tooling, Performance improvements, Formatting/Linting, Security, Technical debt repayment, Release management, Accessibility, Deprecation, Logging/Instrumentation, Internationalization

### Commit Message
Add the blacklist checking to the bulk

### Classification
Bug fixes, New features

### Commit Message
{COMMIT_MESSAGE}
### Classification

---

Figure 5: **GPT-4 1-shot prompt for classifying commits in COMMITPACKFT.**

**xP3x** We use a subset of xP3x (Muennighoff et al., 2022b) focusing on code datasets consisting of APPS (Hendrycks et al., 2021), CodeContests (Li et al., 2022b), Jupyter Code Pairs,[6] MBPP (Austin et al., 2021), XLCoST (Zhu et al., 2022), Code Complex (Jeon et al., 2022), Docstring Corpus (Barone & Sennrich, 2017), Great Code (Hellendoorn et al., 2019) and State Changes.[7]

**OASST** We reuse a filtered variant of OASST (Köpf et al., 2023) from prior work (Dettmers et al., 2023) and apply additional filters to remove responses that refuse to comply with the user request. To compute the programming languages and code fraction for OASST depicted in Table 1, we count all responses containing e.g. ```` ```python ```` or ```` ```py ```` for the Python programming language. There are code samples that are not enclosed in backticks or do not specify the language, thus we are likely underestimating the actual fraction of code data for OASST in Table 1.

## E  COMPARING DATA BEFORE AND AFTER FILTERING

In Table 8 we compare word statistics prior to and after filtering COMMITPACK to create COMMIT-PACKFT. The mean commit subject and message length increases suggesting that messages are more informative in COMMITPACKFT. The code lengths decrease significantly as we limit the number of allowed tokens in the filters in Table 6. This is intended, as we would like to maximize the amount of training signal per token. The code before and after the commit are usually largely the same. By filtering for short samples, we ensure that there are more differences between the code before and after, thus making the model learn faster. The percentage of code changed between pre- and post-commit is $77.6/59.1 = 1.31$ (a 31% increase) as opposed to $3269.8/3269.9 = 1.007$ (a 0.7% increase). Thus, the filtered data carries significantly more signal per token with fewer repetitions of the code prior to the commit.

| Metric | Before Filter | After Filter | Difference |
|---|---|---|---|
| Subject Length (words) | 5.7±0.02 | 6.9±0.01 | +1.28 |
| Message Length (words) | 8.7±0.06 | 9.9±0.05 | +1.34 |
| Pre-Commit Code Length (words) | 3269.9±298.8 | 59.1±0.19 | -3210.9 |
| Post-Commit Code Length (words) | 3269.8±299.5 | 77.6±0.23 | -3214.2 |

Table 8: **The effect of data filters on subject, message, and code lengths**. We compare differences in word statistics of COMMITPACK and COMMITPACKFT.

## F  COMPARING COMMITPACK AND THE STACK

In Table 9 we provide statistics on repositories and usernames of COMMITPACK and The Stack (Kocetkov et al., 2022). COMMITPACK contains a total of 1,934,255 repositories. Around half (49.3%) of them are also in The Stack. However, The Stack only provides the raw code files of these repositories from some fixed point in time. COMMITPACK contains the changes made to the code files in the form of commits. Thus, the same code file may appear multiple times in COMMITPACK for each change that was made to it. Therefore, The Stack only contains 3 terabytes of data, while COMMITPACK contains close to 4.

| Statistic (↓) | COMMITPACK | The Stack 1.2 | Shared | Shared (%) |
|---|---|---|---|---|
| Repositories | 1,934,255 | 18,712,378 | 954,135 | 49.3% |
| Usernames | 825,885 | 6,434,196 | 663,050 | 80.3% |

Table 9: **Overlap in repositories and usernames of COMMITPACK and The Stack.**

---

[6]https://hf.co/datasets/codeparrot/github-jupyter-text-code-pairs
[7]https://hf.co/datasets/Fraser/python-state-changes

## G  PRETRAINING ON COMMITPACK

Due to the scale of COMMITPACK, it is also adequate as a large-scale pretraining dataset. We have included parts of COMMITPACK during the pretraining of StarCoder (Li et al., 2023b) in the format of `<commit_before>code_before<commit_msg>message<commit_after>` `code_after`. We also pretrain a new model, named SANTACODERPACK, with the same architecture as SantaCoder (Allal et al., 2023) on COMMITPACK using this format. We filter COMMITPACK for our six evaluation languages and samples that fit within 8192 tokens leaving us a total of 35B tokens. Following prior work (Muennighoff et al., 2023), we train on this data repeated close to 4 times for a total of 131B tokens taking 14 days. Detailed hyperparameters are in Appendix P.

In Table 10, we benchmark StarCoder and SANTACODERPACK on HUMANEVALFIX using the above-detailed commit format. We find that the commit format leads to very strong performance for StarCoder often surpassing the instruction tuned OCTOCODER from Table 2. However, this pretraining format is not suitable for HUMANEVALEXPLAIN limiting its universality. For SANTACODERPACK, we find performance comparable to SantaCoder, including checkpoints at 131B and 236B tokens. SANTACODERPACK performs slightly worse on Python than SantaCoder. We hypothesize that this discrepancy is due to a *multilingual tax*, as SANTACODERPACK needs to accommodate three additional coding languages (Go, C++ and Rust). SantaCoder has thus more capacity allocated to Python, JavaScript, and Java.

SANTACODERPACK may also be bottlenecked by its small model size of 1.1B parameters. More research into what exactly happens during pretraining (Xia et al., 2022; Biderman et al., 2023a) and how to unify pretraining and instruction tuning are needed. Prior work has also found that including raw code data during pretraining benefits some natural language tasks (Muennighoff et al., 2023; Soldaini et al., 2024; Groeneveld et al., 2024). Future work may consider the effects of including code commit data on natural language tasks.

| Model (↓) | Python | JavaScript | Java | Go | C++ | Rust | Avg. |
|---|---|---|---|---|---|---|---|
| SantaCoder (131B tokens) Instruct Format | 6.5 | 4.2 | 2.9 | - | - | - | - |
| SantaCoder (236B tokens) Instruct Format | 7.1 | 4.2 | 1.8 | - | - | - | - |
| SANTACODERPACK (131B tokens) Commit Format | 3.2 | 4.9 | 1.8 | 3.6 | 4.2 | 1.7 | 3.3 |
| StarCoder Commit Format | 32.7 | 33.6 | 33.0 | 31.9 | 31.6 | 20.2 | 30.5 |

Table 10: **Zero-shot pass@1 (%) performance on HUMANEVALFIX of pretraining experiments.**

## H  HUMANEVALPACK STATISTICS

Table 11 displays statistics of HUMANEVALPACK. Docstrings are largely the same across languages leading to similar statistics except for Rust. As Rust is already a very verbose language as seen by its maximum solution length in Table 11, we do not include examples of how to call the function at the end of its docstrings (see Python docstrings with examples in e.g. Figure 11). Rust also has type annotations for every function so providing these examples is not as needed as it is for e.g. JavaScript which lacks type annotations.

| Statistic (↓) | Python | JavaScript | Java | Go | C++ | Rust |
|---|---|---|---|---|---|---|
| Docstring Avg. Length (chars) | 354 | 352 | 357 | 365 | 352 | 231 |
| Docstring Min. Length (chars) | 56 | 56 | 56 | 56 | 56 | 23 |
| Docstring Max. Length (chars) | 1207 | 1207 | 1211 | 1207 | 1207 | 1067 |
| Solution Avg. Length (chars) | 181 | 234 | 259 | 354 | 295 | 339 |
| Solution Min. Length (chars) | 16 | 19 | 18 | 29 | 17 | 17 |
| Solution Max. Length (chars) | 864 | 1325 | 1399 | 1333 | 1144 | 2157 |

Table 11: **Statistics of HUMANEVALPACK computed across the 164 samples for each language.**

## I    FULL INSTRUCTION DATA ABLATIONS

We provide tabular results of the ablations from Figure 4 in Table 12. We try some additional mixtures, however, none of them perform better than COM-MITPACKFT + OASST. We experiment with changing the formatting to be `<commit_before>old code<commit_msg>message<commit_after>new code` for COMMITPACKFT and `<commit_before><commit_msg>input<commit_after>output` for OASST referred to as the "Formatting" ablation. We hypothesized that aligning the formatting during instruction tuning with the commit format that we used during pretraining (Appendix G) would improve performance. While it seems to improve performance for HUMANEVALFIX compared to our default formatting (see Figure 18), it reduces performance on the other tasks leading to a worse average score of 35.3 in Table 12. "Target Loss" refers to an ablation where we mask loss for inputs as is commonly done during instruction tuning (Muennighoff et al., 2022b). While this leads to the best performance on HUMANEVALSYNTHESIZE, its average performance is worse compared to COMMITPACKFT + OASST, where the loss is computed over the full sequence. We also perform an ablation where we manually select 1178 high-quality samples (725 from OASST and 89, 61, 86, 72, 70 and 75 from COMMITPACKFT for Python, JavaScript, Java, Go, C++ and Rust, respectively). However, this manual selection did not outperform random selection for OCTOCODER. It performed better for OCTOGEEX, however, hence we used it for OCTOGEEX. We hypothesize that our models could achieve significantly better performance by further improving the quality of the instruction data beyond. This may necessitate very careful human selection of samples and manual editing of the data to ensure a uniform style in the outputs. We leave such explorations to future work.

| **Instruction Tuning Dataset (↓)** | HUMANEVALPACK Python | | | **Average** |
| --- | --- | --- | --- | --- |
| | **Fix** | **Explain** | **Synthesize** | |
| Without instruction tuning | 8.7 | 0.0 | 33.6 | 14.1 |
| Self-Instruct (SI) | 23.6 | 0.6 | 43.0 | 22.2 |
| OASST | 23.1 | 34.5 | 46.4 | 34.7 |
| SI + OASST | 24.9 | 28.7 | 46.2 | 33.3 |
| xP3x + OASST | 28.4 | 28.4 | 45.0 | 33.9 |
| COMMITPACKFT + OASST | 30.4 | **35.1** | 46.2 | **37.2** |
| COMMITPACKFT + OASST (Formatting) | 31.1 | 28.9 | 45.8 | 35.3 |
| COMMITPACKFT + OASST (Target loss) | 29.8 | 31.2 | **47.8** | 36.3 |
| COMMITPACKFT + OASST (Manual) | 27.2 | 29.6 | 45.8 | 34.2 |
| COMMITPACKFT + xP3x + OASST | 30.9 | 29.5 | 45.9 | 35.4 |
| COMMITPACKFT + SI + xP3x + OASST | **31.4** | 33.8 | 46.0 | 37.1 |

Table 12: **Zero-shot pass@1 (%) performance across the Python split of HUMANEVALPACK for StarCoder instruction tuning data ablations.**

## J    LINE DIFF FORMAT FOR FIXING CODE

We finetune SantaCoder to experiment with different formatting strategies for fixing bugs comparing *full code generation* and *code diff generation*. When fixing a code bug, usually only a small part of the code needs to change. Only generating the code diff corresponding to the necessary change can make inference significantly more efficient by avoiding repeated characters in the output generation. We finetune SantaCoder on the Python, Java and JavaScript subset of COMMITPACKFT. We exclude other languages as SantaCoder has only been pretrained on these three languages (Allal et al., 2023).

**Commit Format**    For *full code generation*, we reuse the format that we employed for commits in StarCoder pretraining from Appendix G: `<commit_before>code_before<commit_msg>message<commit_after>code_after`. However, SantaCoder has not seen this format during pretraining and does not have special tokens like StarCoder for the delimiters. Thus, for SantaCoder e.g. `<commit_before>` is tokenized as `['<', 'commit', '_', 'before', '>']`.

**Unified diff format**  For *code diff generation*, a simple solution is using the unified diff format,[8] which is a standard way to display changes between code files in a compact and readable format (Lehman et al., 2022; Jung, 2021; Xu et al., 2022b; Monperrus et al., 2021). We depict an example of this format in Figure 6. However, the unified diff format still requires the model to output several unchanged lines below and after the actual modification. Thus, its efficiency gains are limited and there is still unnecessary duplication of the input.

```python
from typing import List

def has_close_elements(numbers: List[float],
    threshold: float) -> bool:
    for idx, elem in enumerate(numbers):
        for idx2, elem2 in enumerate(numbers)
            :
            if idx != idx2:
                distance = elem - elem2
                if distance < threshold:
                    return True

    return False
```

```python
from typing import List

def has_close_elements(numbers: List[float],
    threshold: float) -> bool:
    for idx, elem in enumerate(numbers):
        for idx2, elem2 in enumerate(numbers)
            :
            if idx != idx2:
                distance = abs(elem - elem2)
                if distance < threshold:
                    return True

    return False
```

```
@@ -4,7 +4,7 @@
        for idx, elem in enumerate(numbers):
            for idx2, elem2 in enumerate(numbers):
                if idx != idx2:
-                   distance = elem - elem2
+                   distance = abs(elem - elem2)
                    if distance < threshold:
                        return True
```

Figure 6: **The first problem from the HUMANEVALFIX Python split and the necessary change to fix the bug in unified diff format**. *Top:* Code with and without the bug from Figure 11. *Bottom:* Necessary change to fix the bug in unified diff format.

```
-   7                 distance = elem - elem2
+   7                 distance = abs(elem - elem2)
```

Figure 7: **The line diff format for the problem from Figure 6.**

**Line diff format**  To address the inefficiencies of the unified diff format, we propose the line diff format for representing code differences. There are two requirements for our format: **(1)** The diff can be unambiguously applied to the code before the commit to generate the code after the commit, and **(2)** the code diff should be as short as possible to maximize efficiency by avoiding the inclusion of unchanged code. In Figure 7, we show how our format addresses these. The line diff format keeps track of each change sequentially line-by-line to ensure the code can be correctly modified. By focusing only on the lines that change, we reduce the number of characters in the diff by 70% compared to the unified diff representation in Figure 6.

Both the unified diff format and our line diff format require the model to predict line numbers. This is very challenging when training on raw code as models need to count and keep track of line numbers. To simplify line number prediction, we automatically add line numbers to the raw code in the finetuning dataset for the line diff format. This allows the model to simply copy the line number into the output simplifying the diff generation. However, it diminishes efficiency slightly by adding additional input tokens that the model needs to process.

As summarized in Table 13, finetuning SantaCoder using the line diff format significantly improves performance compared to prior finetuning on HUMANEVALFIX across all languages. It also outperforms finetuning using the commit format, which only provides gains on JavaScript and Java compared to no finetuning. However, finetuning on the diff format may converge slower than the commit format as the diff format significantly differs from the raw code seen during pretraining.

---

[8]https://en.wikipedia.org/wiki/Diff#Unified_format

Figures 8, 9, 10 show line diff generations of our model. A limitation of our current line diff implementation is that it does not handle code insertion well. The inserted lines may change the line numbers of all following lines, which can result in problems when applying the diff. Further, the diff format is not useful for HUMANEVALEXPLAIN and HUMANEVALSYNTHESIZE. Future work could consider training models that can both be instructed to use the line diff format, such as for HUMANEVALFIX, but also explain or synthesize code without producing a diff.

| Model | Python | JavaScript | Java |
|---|---|---|---|
| SantaCoder | 7.1 | 4.2 | 1.8 |
| SantaCoder + Commit format finetuning | 3.8 | 5.3 | 9.2 |
| SantaCoder + Line diff format finetuning | **9.9** | **9.7** | **10.0** |

Table 13: **Zero-shot pass@1 (%) performance on HUMANEVALFIX of SantaCoder formatting experiments.**

```
-  3      let depth = 0, max_depth = 0;
+  3      let depth = 0, max_depth = 1;
- 12      return max_depth;
+ 12      return max_depth - 1;
- 14   return paren_string.split(' ')
- 15          .filter(x => x != '')
- 16          .map(x => parseParenGroup(x));
- 17 }
+ 14   let paren_list = paren_string.split(' ');
+ 15   let nested_parens = paren_list.map(x => parseParenGroup(x));
+ 16   return nested_parens.reduce((prev, curr) => {
+ 17     if (prev == 0) {
+ 18       return curr;
+ 19     } else {
+ 20       return curr - 1;
+ 21     }
+ 22   });
+ 23 }
```

Figure 8: **A line diff generation of our model on a JavaScript HUMANEVALFIX problem.**

```
- 18              if (current_depth < 0) {
+ 18              if (current_depth < 0 && current_string.length() > 0) {
```

Figure 9: **A line diff generation of our model on a Java HUMANEVALFIX problem.**

```
-  2        for i, l1 in enumerate(l):
-  3            for j in range(i, len(l)):
+  2        for i in range(0, len(l)):
+  3            for j in range(i+1, len(l)):
```

Figure 10: **A line diff generation of our model on a Python HUMANEVALFIX problem.**

## K   RESULTS ON HUMANEVALFIXDOCS

The default version of HUMANEVALFIX does not include docstrings, but only provides the unit tests to the model alongside the buggy function. An alternative is providing docstrings as the source of ground truth for the model to fix the buggy function. Solving from docstrings is generally easier for models than from tests, as models can also solve it via pure code synthesis without looking at the buggy function at all. We provide results of some models on this variant in Table 14. For StarCoder, we distinguish two prompting formats: An instruction to fix bugs like in Figure 3 or the commit format it has seen during pretraining (Appendix G). OCTOCODER performs strongly on this variant. However, directly using StarCoder with the commit format from pretraining (Appendix G) is slightly better. This is in line with the commit format from pretraining also performing slightly better on HUMANEVALFIX in Table 10 compared to OCTOCODER in Table 2. Diff Codegen 2B (Bradley et al., 2023) performs poorly as its predicted code diffs are often irrelevant to the actual bug, see Figure 39.

| Model | Python | JavaScript | Java | Go | C++ | Rust | Avg. |
|---|---|---|---|---|---|---|---|
| Non-permissive models | | | | | | | |
| GPT-4 | 88.4 | 80.5 | 82.9 | 81.1 | 82.3 | 68.9 | 80.7 |
| Permissive Models | | | | | | | |
| Diff Codegen 2B | 0.0 | 0.1 | 0.0 | 0.3 | 0.0 | 0.2 | 0.1 |
| StarCoder Commit Format | **58.8** | **49.2** | 43.9 | **55.2** | **51.5** | **41.8** | **50.1** |
| StarCoder Instruct Format | 41.7 | 30.7 | 44.3 | 34.5 | 28.7 | 14.0 | 26.5 |
| OCTOCODER | 53.8 | 48.1 | **54.3** | 54.9 | 49.2 | 32.1 | 48.7 |

Table 14: **Zero-shot pass@1 (%) performance on HUMANEVALFIXDOCS.**

## L   HUMANEVALFIX BUG TYPES

Table 15 contains an overview of bugs that were manually added by one of the authors to HumanEval solutions for the construction of HUMANEVALFIX. Figures 11-16 contain an example of each type from the Python split. The bug type for each problem is the same across all programming languages in HUMANEVALFIX, but for a few samples it affects a different part of the solution due to the code solutions not being perfectly parallel across languages.

| Bug type | Subtype | Explanation | Example | Count |
|---|---|---|---|---|
| Missing logic | | Misses code needed to solve the problem | Figure 11 | 33 |
| Excess logic | | Contains excess code leading to mistakes | Figure 12 | 31 |
| | Value misuse | An incorrect value is used | Figure 13 | 44 |
| Wrong logic | Operator misuse | An incorrect operator is used | Figure 14 | 25 |
| | Variable misuse | An incorrect variable is used | Figure 15 | 23 |
| | Function misuse | An incorrect function is used | Figure 16 | 8 |
| Total | | | | 164 |

Table 15: **HUMANEVALFIX bug types.**

```python
from typing import List

def has_close_elements(numbers: List[float
    ], threshold: float) -> bool:
    """ Check if in given list of numbers,
        are any two numbers closer to
        each other than
    given threshold.
    >>> has_close_elements([1.0, 2.0,
        3.0], 0.5)
    False
    >>> has_close_elements([1.0, 2.8, 3.0,
        4.0, 5.0, 2.0], 0.3)
    True
    """
    for idx, elem in enumerate(numbers):
        for idx2, elem2 in enumerate(
            numbers):
            if idx != idx2:
                distance = abs(elem -
                    elem2)
                if distance < threshold:
                    return True

    return False
```

```python
from typing import List

def has_close_elements(numbers: List[float
    ], threshold: float) -> bool:
    """ Check if in given list of numbers,
        are any two numbers closer to
        each other than
    given threshold.
    >>> has_close_elements([1.0, 2.0,
        3.0], 0.5)
    False
    >>> has_close_elements([1.0, 2.8, 3.0,
        4.0, 5.0, 2.0], 0.3)
    True
    """
    for idx, elem in enumerate(numbers):
        for idx2, elem2 in enumerate(
            numbers):
            if idx != idx2:
                distance = elem - elem2
                if distance < threshold:
                    return True

    return False
```

Figure 11: **Missing logic bug example.** The buggy code (*right*) misses the 'abs' statement.

```python
def truncate_number(number: float) ->
    float:
    """ Given a positive floating point
        number, it can be decomposed into
    and integer part (largest integer
        smaller than given number) and
        decimals
    (leftover part always smaller than 1).

    Return the decimal part of the number.
    >>> truncate_number(3.5)
    0.5
    """
    return number % 1.0
```

```python
def truncate_number(number: float) ->
    float:
    """ Given a positive floating point
        number, it can be decomposed into
    and integer part (largest integer
        smaller than given number) and
        decimals
    (leftover part always smaller than 1).

    Return the decimal part of the number.
    >>> truncate_number(3.5)
    0.5
    """
    return number % 1.0 + 1.0
```

Figure 12: **Excess logic bug example.** The buggy code (*right*) incorrectly adds 1 to the result.

```python
from typing import List, Tuple

def sum_product(numbers: List[int]) ->
    Tuple[int, int]:
    """ For a given list of integers,
        return a tuple consisting of a
        sum and a product of all the
        integers in a list.
    Empty sum should be equal to 0 and
        empty product should be equal to
        1.
    >>> sum_product([])
    (0, 1)
    >>> sum_product([1, 2, 3, 4])
    (10, 24)
    """
    sum_value = 0
    prod_value = 1

    for n in numbers:
        sum_value += n
        prod_value *= n
    return sum_value, prod_value
```

```python
from typing import List, Tuple

def sum_product(numbers: List[int]) ->
    Tuple[int, int]:
    """ For a given list of integers,
        return a tuple consisting of a
        sum and a product of all the
        integers in a list.
    Empty sum should be equal to 0 and
        empty product should be equal to
        1.
    >>> sum_product([])
    (0, 1)
    >>> sum_product([1, 2, 3, 4])
    (10, 24)
    """
    sum_value = 0
    prod_value = 0

    for n in numbers:
        sum_value += n
        prod_value *= n
    return sum_value, prod_value
```

Figure 13: **Value misuse bug example.** The buggy code (*right*) incorrectly initializes the product to 0.

```python
from typing import List

def below_zero(operations: List[int]) ->
    bool:
    """ You're given a list of deposit and
        withdrawal operations on a bank
        account that starts with
    zero balance. Your task is to detect
        if at any point the balance of
        account fallls below zero, and
    at that point function should return
        True. Otherwise it should return
        False.
    >>> below_zero([1, 2, 3])
    False
    >>> below_zero([1, 2, -4, 5])
    True
    """
    balance = 0

    for op in operations:
        balance += op
        if balance < 0:
            return True

    return False
```

```python
from typing import List

def below_zero(operations: List[int]) ->
    bool:
    """ You're given a list of deposit and
        withdrawal operations on a bank
        account that starts with
    zero balance. Your task is to detect
        if at any point the balance of
        account fallls below zero, and
    at that point function should return
        True. Otherwise it should return
        False.
    >>> below_zero([1, 2, 3])
    False
    >>> below_zero([1, 2, -4, 5])
    True
    """
    balance = 0

    for op in operations:
        balance += op
        if balance == 0:
            return True

    return False
```

Figure 14: **Operator misuse bug example.** The buggy code (*right*) incorrectly checks for equality with 0.

```python
from typing import List

def mean_absolute_deviation(numbers: List[
    float]) -> float:
    """ For a given list of input numbers,
        calculate Mean Absolute
        Deviation
    around the mean of this dataset.
    Mean Absolute Deviation is the average
        absolute difference between each
    element and a centerpoint (mean in
        this case):
    MAD = average | x - x_mean |
    >>> mean_absolute_deviation([1.0, 2.0,
        3.0, 4.0])
    1.0
    """
    mean = sum(numbers) / len(numbers)
    return sum(abs(x - mean) for x in
        numbers) / len(numbers)
```

```python
from typing import List

def mean_absolute_deviation(numbers: List[
    float]) -> float:
    """ For a given list of input numbers,
        calculate Mean Absolute
        Deviation
    around the mean of this dataset.
    Mean Absolute Deviation is the average
        absolute difference between each
    element and a centerpoint (mean in
        this case):
    MAD = average | x - x_mean |
    >>> mean_absolute_deviation([1.0, 2.0,
        3.0, 4.0])
    1.0
    """
    mean = sum(numbers) / len(numbers)
    return sum(abs(x - mean) for x in
        numbers) / mean
```

Figure 15: **Variable misuse bug example.** The buggy code (*right*) incorrectly divides by the mean.

```python
def flip_case(string: str) -> str:
    """ For a given string, flip lowercase
        characters to uppercase and
        uppercase to lowercase.
    >>> flip_case('Hello')
    'hELLO'
    """
    return string.swapcase()
```

```python
def flip_case(string: str) -> str:
    """ For a given string, flip lowercase
        characters to uppercase and
        uppercase to lowercase.
    >>> flip_case('Hello')
    'hELLO'
    """
    return string.lower()
```

Figure 16: **Function misuse bug example.** The buggy code (*right*) incorrectly uses the 'lower()' function.

## M   PERFORMANCE BREAKDOWN BY HUMANEVALFIX BUG TYPE

All bugs in HUMANEVALFIX are categorized into bug types as described in Appendix L. In Table 16, we break down the HUMANEVALFIX performance of select models from Table 2 by bug type. We find that models struggle most with bugs that require removing excess logic (e.g. Figure 12). WizardCoder is only able to solve 11% of excess logic bugs while solving about four times more bugs that relate to value misuse. The performance of OCTOGEEX and OCTOCODER is more stable than WizardCoder across the different bug types, possibly due to the diversity of COMMITPACKFT as displayed in Figure 2. GPT-4 performs best across all bug types.

| Bug type | Subtype | OCTOGEEX | OCTOCODER | WizardCoder | GPT-4 |
|---|---|---|---|---|---|
| Missing logic | | 24.2 | 24.4 | 31.2 | 45.5 |
| Excess logic | | 16.3 | 16.9 | 11.0 | 38.7 |
| | Value misuse | 33.2 | 34.7 | 45.1 | 50.0 |
| | Operator misuse | 32.8 | 42.0 | 34.4 | 56.0 |
| Wrong logic | Variable misuse | 35.7 | 33.7 | 30.4 | 43.5 |
| | Function misuse | 25.0 | 37.5 | 37.5 | 50.0 |
| Overall | | 28.1 | 30.4 | 31.8 | 47.0 |

Table 16: **Breakdown of HUMANEVALFIX Python pass@1 (%) performance by bug type for select models.** Statistics for each bug type are in Table 15.

## N   HUMANEVALEXPLAIN WITH FILL-IN-THE-MIDDLE

In Table 2, all models are prompted with the same instruction to provide an explanation (but using slightly different formats, see Appendix Q). For StarCoder, we can alternatively prompt it with the Fill-in-the-Middle (FIM) technique (Bavarian et al., 2022), which it already knows from pretraining (Li et al., 2023b). To do so, we provide it with a prompt as shown in Figure 17 to generate the docstring. While a docstring is not necessarily an explanation, it can be similar. The results in Table 17 show that it performs significantly better with this prompting strategy than in Table 2. However, it still falls short of OCTOCODER and other models likely due to the imperfect approximation of an explanation.

```
<fim_prefix>from typing import List

def has_close_elements(numbers: List[float], threshold: float) -> bool:
    """ <fim_suffix>
    """
    for idx, elem in enumerate(numbers):
        for idx2, elem2 in enumerate(numbers):
            if idx != idx2:
                distance = abs(elem - elem2)
                if distance < threshold:
                    return True

    return False<fim_middle>
```

Figure 17: **FIM prompt example for StarCoder.**

| Model | Python | JavaScript | Java | Go | C++ | Rust | Avg. |
|---|---|---|---|---|---|---|---|
| StarCoder FIM | 19.4 | 17.6 | 16.3 | 11.8 | 17.9 | 16.7 | 16.6 |

Table 17: **Performance of StarCoder on HUMANEVALEXPLAIN with FIM.**

## O   HUMANEVALEXPLAIN BLEU AND METEOR COMPARISON

By default, we use pass@$k$ to evaluate HUMANEVALEXPLAIN (§3). In Table 18 we compare it with BLEU (Papineni et al., 2002) and METEOR (Banerjee & Lavie, 2005). While our pass@$k$ formulation does not require a ground truth explanation, BLEU and METEOR do. This can be a limiting factor. For this evaluation, we use the function docstrings as the ground-truth explanation to compute the BLEU and METEOR scores. We compute BLEU and METEOR for each of the $n = 20$ generations (3) and select the highest score. The scores are then averaged across the 164 samples for each language. Rust scores are the highest which is likely due to Rust docstrings containing no example function calls (Appendix H).

| Metric ($\downarrow$) | Python | JavaScript | Java | Go | C++ | Rust | Avg. |
|---|---|---|---|---|---|---|---|
| pass@1 | 35.1 | 24.5 | 27.3 | 21.1 | 24.1 | 14.8 | 24.5 |
| BLEU-1 | 7.1 | 7.0 | 6.3 | 6.1 | 6.4 | 8.1 | 6.8 |
| BLEU-2/3/4 | 0.0 | 0.0 | 0.0 | 0.0 | 0.0 | 0.0 | 0.0 |
| METEOR | 7.8 | 7.6 | 7.2 | 7.4 | 7.0 | 9.4 | 7.7 |

Table 18: **Comparison of different metrics for HUMANEVALEXPLAIN on explanations by OCTOCODER.** Pass@1 is computed with respect to a generated solution given the explanation (§3) while BLEU and METEOR are computed by comparing the explanation with the docstring.

## P   HYPERPARAMETERS

**StarCoder finetuning (OCTOCODER)**   For all experiments finetuning StarCoder, we use a learning rate of 5e-4 with a cosine schedule and linear warmup. We use a batch size of 32 and train for up to one epoch, as we did not observe benefits from more steps. OCTOCODER was trained for 35 steps with a sequence length of 2048 and packing corresponding to 2.2 million total finetuning tokens. We use LoRA (Hu et al., 2021) as we did not observe a significant difference from full finetuning. Follow-up work has further investigated this choice (Zhuo et al., 2024).

**CodeGeeX finetuning (OCTOGEEX)**   To create OCTOGEEX, we finetune CodeGeeX2 for 35 steps with a batch size of 48 and a learning rate of 5e-5 largely following the OCTOCODER setup.

**SantaCoder finetuning**   For all experiments finetuning SantaCoder, we use a learning rate of 5e-5 with a cosine schedule and linear warmup. We finetune SantaCoder using a batch size of 64 for up to 200,000 steps.

**SantaCoder pretraining (SANTACODERPACK)**   We follow the setup from Allal et al. (2023) to pretrain on COMMITPACK except for using a sequence length of 8192 and the StarCoder tokenizer, which has special tokens for the commit format delimiters (see Appendix G). SANTACODERPACK utilizes Multi Query Attention (MQA) (Shazeer, 2019) but removes Fill-in-the-Middle (FIM) (Bavarian et al., 2022). We conducted pretraining on 32 A100 GPUs, totaling 250k training steps, with a global batch size of 64. Other hyperparameter settings follow SantaCoder, including using Adam with $\beta_1 = 0.9, \beta_2 = 0.95, \epsilon = 10^{-8}$, and a weight decay of 0.1. The learning rate is set to $2 \times 10^{-4}$ and follows a cosine decay after warming up for 2% of the training steps.

## Q   PROMPTS

The prompting format can significantly impact performance. In the spirit of true few-shot learning (Perez et al., 2021) we do not optimize prompts and go with the format provided by the respective model authors or the most intuitive format if none is provided. For each task, we define an instruction, an optional context and an optional function start (Table 19). The function start is provided to make sure the model directly completes the function without having to search for the function in the model output. These three parts are then combined in slightly different ways for each model (Figures 18-24). We implement our evaluation using open-source frameworks (Ben Allal et al., 2022; Gao et al., 2021).

| HUMANEVALFIX | |
|---|---|
| Instruction | Fix bugs in has_close_elements. |
| Context | ```
from typing import List

def has_close_elements(numbers: List[float], threshold: float) -> bool:
    for idx, elem in enumerate(numbers):
        for idx2, elem2 in enumerate(numbers):
            if idx != idx2:
                distance = elem - elem2
                if distance < threshold:
                    return True

    return False
``` |
| Function start | ```
from typing import List

def has_close_elements(numbers: List[float], threshold: float) -> bool:
``` |
| **HUMANEVALEXPLAIN** | |
| Instruction (Describe) | Provide a concise natural language description of the code using at most 213 characters. |
| Context (Describe) | ```
from typing import List

def has_close_elements(numbers: List[float], threshold: float) -> bool:
    for idx, elem in enumerate(numbers):
        for idx2, elem2 in enumerate(numbers):
            if idx != idx2:
                distance = abs(elem - elem2)
                if distance < threshold:
                    return True

    return False
``` |
| Instruction (Synthesize) | Write functional code in Python according to the description. |
| Context (Synthesize) | {Description generated by the model} |
| Function start (Synthesize) | ```
from typing import List

def has_close_elements(numbers: List[float], threshold: float) -> bool:
``` |
| **HUMANEVALSYNTHESIZE** | |
| Instruction | Write a Python function 'has_close_elements(numbers: List[float], threshold: float) -> bool' to solve the following problem:
Check if in given list of numbers, are any two numbers closer to each other than given threshold.
»> has_close_elements([1.0, 2.0, 3.0], 0.5)
False
»> has_close_elements([1.0, 2.8, 3.0, 4.0, 5.0, 2.0], 0.3)
True |
| Function start | ```
from typing import List

def has_close_elements(numbers: List[float], threshold: float) -> bool:
    """ Check if in given list of numbers, are any two numbers closer
        to each other than given threshold.
    >>> has_close_elements([1.0, 2.0, 3.0], 0.5)
    False
    >>> has_close_elements([1.0, 2.8, 3.0, 4.0, 5.0, 2.0], 0.3)
    True
    """
``` |

Table 19: **Instructions and function examples used.** If no function start or no context is present, that part is not added to the prompt (and the preceding newline is also removed).

---

Question: {instruction}
{context}

Answer:
{function_start}

---

Figure 18: **OCTOCODER and OCTOGEEX prompting format**

---

Below is an instruction that describes a task. Write a response that appropriately completes the request.

### Instruction:
{instruction}
{context}

### Response:
{function_start}

---

Figure 19: **WizardCoder prompting format from their codebase.**[9]

---

Below is an instruction that describes a task. Write a response that appropriately completes the request.

### Instruction:
{instruction}
{context}

### Response:{function_start}

---

Figure 20: **InstructCodeT5+ decoder prompting format from their codebase.**[10] The encoder receives the same prompt without the function start.

---

[9] https://github.com/nlpxucan/WizardLM/blob/9c6fb34d012d60dc4f31348ee0a8e35335c04564/WizardCoder/src/humaneval_gen.py#L38

[10] https://github.com/salesforce/CodeT5/blob/main/CodeT5%2B/humaneval/generate_codet5p.py#L89

[11] https://huggingface.co/HuggingFaceH4/starchat-beta

```
<|system|>
<|end|>
<|user|>
{instruction}
{context}<|end|>
<|assistant|>
{function_start}
```

Figure 21: **StarChat-$\beta$ prompting format from their documentation.**[11]

```
{context}
{instruction}
{function_start}
```

Figure 22: **Default prompting format (StarCoder, BLOOMZ).** Used for Figure 3.

```
{context}
{instruction}
```

Figure 23: **GPT-4 prompting format.** Same as Figure 22 but excluding the function start, as we found the model would not continue from the provided function start, likely due to the chat mode.

```
{context}
{instruction}
 Start your code with:
{func_start}
```

Figure 24: **GPT-4 prompting format for HUMANEVALEXPLAIN (Synthesize).** We include the function start for the synthesize part of HUMANEVALEXPLAIN in case the function name is not included in the code description generated by GPT-4.

```
<NME> {filename}
<BEF> {context}
<MSG> {instruction}
<DFF>
```

Figure 25: **Prompting format for Diff Codegen 2B evaluated in Appendix K.**

# R EXAMPLES

## R.1 OCTOCODER

*Input:*

Question: Fix bugs in sum_product.

```c++
#include<stdio.h>
#include<vector>
using namespace std;
#include<algorithm>
#include<math.h>
#include<stdlib.h>
vector<int> sum_product(vector<int> numbers){
    int sum=0,product=0;
    for (int i=0;i<numbers.size();i++)
    {
        sum+=numbers[i];
        product*=numbers[i];
    }
    return {sum,product};
}
```

Answer:

```c++
#include<stdio.h>
#include<vector>
using namespace std;
#include<algorithm>
#include<math.h>
#include<stdlib.h>
vector<int> sum_product(vector<int> numbers){
```

OCTOCODER:

```c++
    int sum=0,product=1;
    for (int i=0;i<numbers.size();i++)
    {
        sum+=numbers[i];
        product*=numbers[i];
    }
    return {sum,product};
}
```

Figure 26: **Correct solution from OCTOCODER to a HUMANEVALFIX C++ problem.** The corresponding Python problem is depicted in Figure 13.

*Input:*

Question: Write a Java function 'public String decodeShift(String s)' to solve the following problem:
takes as input string encoded with encodeShift function. Returns decoded string.
Answer:

```java
import java.util.*;
import java.lang.*;

class Solution {
    /**
    returns encoded string by shifting every character by 5 in the alphabet.
     */
    public String encodeShift(String s) {
        StringBuilder sb = new StringBuilder();
        for (char ch : s.toCharArray()) {
            sb.append((char) ('a' + ((ch + 5 - 'a') % 26)));
        }
        return sb.toString();
    }

    /**
    takes as input string encoded with encodeShift function. Returns decoded string.
     */
    public String decodeShift(String s) {
```

OCTOCODER*:*

```java
        StringBuilder sb = new StringBuilder();
        for (char ch : s.toCharArray()) {
            sb.append((char) ('a' + ((ch - 'a' - 5) % 26)));
        }
        return sb.toString();
    }
}
```

Figure 27: **Incorrect solution from OCTOCODER to a HUMANEVALSYNTHESIZE Java problem.**
The correct solution would be to add 21 to ch instead of subtracting 5 from it.

*Input:*

Question: Provide a concise natural language description of the code using at most 671 characters.

```go
func FruitDistribution(s string,n int) int {
    split := strings.Split(s, " ")
        for _, i := range split {
                atoi, err := strconv.Atoi(i)
                if err != nil {
                        continue
                }
                n = n - atoi
        }
        return n
}
```

Answer:

OCTOCODER*:*

The function FruitDistribution takes two arguments, a string s and an integer n. The function splits the string s into a slice of strings using the space character as the delimiter. The function iterates over the slice and converts each string to an integer using the strconv.Atoi function. If the conversion is successful, the integer is subtracted from n. The function returns the value of n after all conversions have been performed.

*Input (without previous input):*

Question: Write functional code in Go according to the description.
The function FruitDistribution takes two arguments, a string s and an integer n. The function splits the string s into a slice of strings using the space character as the delimiter. The function iterates over the slice and converts each string to an integer using the strconv.Atoi function. If the conversion is successful, the integer is subtracted from n. The function returns the value of n after all conversions have been performed.
Answer:

```go
func FruitDistribution(s string,n int) int {
```

OCTOCODER*:*

```go
        fruits := strings.Split(s, " ")
        for _, fruit := range fruits {
                if num, err := strconv.Atoi(fruit); err == nil {
                        n -= num
                }
        }
        return n
}
```

Figure 28: **Correct solution from OCTOCODER to a HUMANEVALEXPLAIN Go problem.** Given its own description OCTOCODER writes a correct solution to the problem. Its solution is more compact and efficient than the original solution.

## R.2  GPT-4

*User:*

Write a Go function 'func RescaleToUnit(numbers []float64) []float64' to solve the following problem:
Given a list of numbers (of at least two elements), apply a linear transform to that list,
such that the smallest number will become 0 and the largest will become 1
>>> RescaleToUnit([1.0, 2.0, 3.0, 4.0, 5.0])
[0.0, 0.25, 0.5, 0.75, 1.0]

*GPT-4:*

Here is a Go function that solves the problem:

```go
package main

import (
    "fmt"
    "math"
)

func RescaleToUnit(numbers []float64) []float64 {
    min := numbers[0]
    max := numbers[0]

    // Find the min and max values in the list
    for _, number := range numbers {
        if number < min {
            min = number
        }
        if number > max {
            max = number
        }
    }

    // Apply the linear transform to each number in the list
    for i, number := range numbers {
        numbers[i] = (number - min) / (max - min)
    }

    return numbers
}

func main() {
    fmt.Println(RescaleToUnit([]float64{1.0, 2.0, 3.0, 4.0, 5.0}))
}
```

This function first finds the minimum and maximum values in the list of numbers. Then it applies the linear transformation to each number in the list. The transformation is '(number - min) / (max - min)', which scales the number such that 'min' becomes 0 and 'max' becomes 1. The transformed numbers are returned in a new list.

Figure 29: **Incorrect solution from GPT-4 to a HUMANEVALSYNTHESIZE Go problem.** GPT-4 includes an import that it does not use (math) which leads to a compilation error in Golang and thus the function does not pass the test cases.

*User:*

Provide a concise natural language description of the code using at most 246 characters.

```python
def truncate_number(number: float) -> float:
    return number % 1.0
```

*GPT-4:*

The code defines a function that takes a float as input and returns its fractional part by using the modulus operator with 1.0.

*User (without previous conversation history):*

The code defines a function that takes a float as input and returns its fractional part by using the modulus operator with 1.0.
Write functional code in Python according to the description.
Start your code with:

```python
def truncate_number(number: float) -> float:
```

*GPT-4:*

```python
def truncate_number(number: float) -> float:
    return number % 1.0
```

Figure 30: **Correct solution from GPT-4 to a HUMANEVALEXPLAIN Python problem.** GPT-4 writes a description that is significantly shorter than the docstring and thus within the character limit. It then generates the perfect solution from its own description. The docstring for this problem can be seen in Figure 12.

*User:*

```rust
fn has_close_elements(numbers:Vec<f32>, threshold: f32) -> bool{
    for i in 0..numbers.len(){
        for j in 1..numbers.len(){

            if i != j {
                let distance:f32 = numbers[i] - numbers[j];

            if distance < threshold{
                return true;
            }

            }

        }
    }

    return false;

}
```

```python
def check(has_close_elements):
    assert has_close_elements([1.0, 2.0, 3.9, 4.0, 5.0, 2.2], 0.3) == True
    assert has_close_elements([1.0, 2.0, 3.9, 4.0, 5.0, 2.2], 0.05) == False
    assert has_close_elements([1.0, 2.0, 5.9, 4.0, 5.0], 0.95) == True
    assert has_close_elements([1.0, 2.0, 5.9, 4.0, 5.0], 0.8) == False
    assert has_close_elements([1.0, 2.0, 3.0, 4.0, 5.0, 2.0], 0.1) == True
    assert has_close_elements([1.1, 2.2, 3.1, 4.1, 5.1], 1.0) == True
    assert has_close_elements([1.1, 2.2, 3.1, 4.1, 5.1], 0.5) == False

check(has_close_elements)
```

Fix bugs in has_close_elements.

*GPT-4:*

The function has_close_elements is supposed to check if there are any two elements in the vector that are closer than a given threshold. However, there are a few issues with the current implementation:
1. The distance calculation is not correct. It should be the absolute difference between the two numbers, not just the difference.
2. The second loop should start from i+1, not 1. This is because we only need to compare each number with the numbers after it, not the numbers before it.
Here is the corrected function:

```rust
fn has_close_elements(numbers:Vec<f32>, threshold: f32) -> bool{
    for i in 0..numbers.len(){
        for j in i+1..numbers.len(){
            let distance:f32 = (numbers[i] - numbers[j]).abs();
            if distance < threshold{
                return true;
            }
        }
    }
    return false;
}
```

This function will now correctly check if there are any two elements in the vector that are closer than the given threshold.

Figure 31: **Correct solution from GPT-4 to a HUMANEVALFIX Rust problem.** GPT-4 correctly fixes the bug by adding the necessary 'abs' statement to compute the absolute value. It additionally improves the efficiency of the function by starting the inner loop from 'i+1' instead of 1 like in the original solution. As some problems in HUMANEVALPACK are not perfectly efficient, evaluating the ability of models to improve the efficiency of the solutions is another interesting benchmarking direction (Yetistiren et al., 2022).

*User:*

```python
def car_race_collision(n: int):
    return n**3

def check(car_race_collision):
    assert car_race_collision(2) == 4
    assert car_race_collision(3) == 9
    assert car_race_collision(4) == 16
    assert car_race_collision(8) == 64
    assert car_race_collision(10) == 100

check(car_race_collision)
```

Fix bugs in car_race_collision.

*GPT-4:*

The function car_race_collision seems to be working fine as it is. It takes an integer 'n' as input and returns the cube of 'n'. There don't seem to be any bugs in this function.

However, if the function is supposed to calculate something specific related to a car race collision and it's not doing that correctly, we would need more information about the expected behavior to fix it.

Figure 32: **Incorrect solution from GPT-4 to a HUMANEVALFIX Python problem.** GPT-4 does not catch that the test cases expect the function to square, not cube n. OCTOCODER correctly modifies the 3 to 2 in 6 out of $n = 20$ generations.

---

*User:*

---

```python
def rounded_avg(n, m):
    if m < n:
        return -1
    summation = 0
    for i in range(n, m+1):
        summation += i
    return bin(round(summation/(m - n)))

def check(rounded_avg):

    # Check some simple cases
    assert rounded_avg(1, 5) == "0b11"
    assert rounded_avg(7, 13) == "0b1010"
    assert rounded_avg(964,977) == "0b1111001010"
    assert rounded_avg(996,997) == "0b1111100100"
    assert rounded_avg(560,851) == "0b1011000010"
    assert rounded_avg(185,546) == "0b101101110"
    assert rounded_avg(362,496) == "0b110101101"
    assert rounded_avg(350,902) == "0b1001110010"
    assert rounded_avg(197,233) == "0b11010111"

    # Check some edge cases that are easy to work out by hand.
    assert rounded_avg(7, 5) == -1
    assert rounded_avg(5, 1) == -1
    assert rounded_avg(5, 5) == "0b101"

check(rounded_avg)
```

Fix bugs in rounded_avg.

---

*GPT-4:*

---

The function rounded_avg is supposed to calculate the average of all numbers between n and m (inclusive), round it, and then return the binary representation of the result.

There are several bugs in the function:

1. The function should return -1 if m < n, but it should also return -1 if m == n because in this case, the denominator of the average calculation would be 0, which would cause a division by zero error.

2. The function calculates the average by summing all numbers between n and m and then dividing by (m - n). However, this is incorrect because the number of integers between n and m (inclusive) is (m - n + 1), not (m - n).

3. The function uses the bin function to convert the result to binary. However, the bin function returns a string that starts with '0b' to indicate that it is a binary number. If the function is supposed to return the binary representation as an integer, it should remove this prefix.

Here is the corrected function:

```python
def rounded_avg(n, m):
    if m <= n:
        return -1
    summation = 0
    for i in range(n, m+1):
        summation += i
    average = round(summation / (m - n + 1))
    return int(bin(average)[2:])
```

This function now correctly calculates the average, rounds it, and returns the binary representation as an integer. If n is greater than or equal to m, it returns -1.

---

Figure 33: **Incorrect solution from GPT-4 to a HUMANEVALFIX Python problem.** The bug consists of a "+1" missing after "m-n". GPT-4 correctly fixes that, however, it introduces two new bugs: Retuning "-1" when m=n and removing the "0b", both of which evidently lead to the provided unit tests failing.

### R.3 WIZARDCODER

---

*Input:*

---

Below is an instruction that describes a task. Write a response that appropriately completes the request.
### Instruction: Fix bugs in bf.

```
def bf(planet1, planet2):
    planet_names = ("Mercury", "Venus", "Earth", "Mars", "Jupyter", "Saturn", "Uranus", "
        Neptune")
    if planet1 not in planet_names or planet2 not in planet_names or planet1 == planet2:
        return ()
    planet1_index = planet_names.index(planet1)
    planet2_index = planet_names.index(planet2)
    if planet1_index < planet2_index:
        return (planet_names[planet1_index + 1: planet2_index])
    else:
        return (planet_names[planet2_index + 1 : planet1_index])

def check(bf):

    # Check some simple cases
    assert bf("Jupiter", "Neptune") == ("Saturn", "Uranus"), "First test error: " + str(len(bf
        ("Jupiter", "Neptune")))
    assert bf("Earth", "Mercury") == ("Venus",), "Second test error: " + str(bf("Earth", "
        Mercury"))
    assert bf("Mercury", "Uranus") == ("Venus", "Earth", "Mars", "Jupiter", "Saturn"), "Third
        test error: " + str(bf("Mercury", "Uranus"))
    assert bf("Neptune", "Venus") == ("Earth", "Mars", "Jupiter", "Saturn", "Uranus"), "Fourth
        test error: " + str(bf("Neptune", "Venus"))

    # Check some edge cases that are easy to work out by hand.
    assert bf("Earth", "Earth") == ()
    assert bf("Mars", "Earth") == ()
    assert bf("Jupiter", "Makemake") == ()

check(bf)
```

### Response:

```
def bf(planet1, planet2):
```

---

*WizardCoder:*

---

```
    planet_names = ("Mercury", "Venus", "Earth", "Mars", "Jupyter", "Saturn", "Uranus", "
        Neptune")
    if planet1 not in planet_names or planet2 not in planet_names or planet1 == planet2:
        return ()
    planet1_index = planet_names.index(planet1)
    planet2_index = planet_names.index(planet2)
    if planet1_index < planet2_index:
        return (planet_names[planet1_index + 1: planet2_index])
    else:
        return (planet_names[planet2_index + 1 : planet1_index])
```

---

Figure 34: **Incorrect solution from WizardCoder to a HUMANEVALFIX Python problem.** The bug consists of "Jupiter" being misspelled as "Jupyter" in the function. WizardCoder returns the same function without making a single change leaving the bug. This is a particularly difficult bug that only GPT-4 and OCTOCODER (only for 1 out of the n=20 generations) are able to solve.

## R.4 BLOOMZ

*Input:*

```
const generateIntegers = (a, b) => {
  if (a > b) {
    let tmp = a;
    a = b;
    b = tmp;
  }
  let y = []
  for (let i = a; i <= b; i++) {
    if (i == 2 || i == 4 || i == 6 || i == 8) { y.push(i) }
  }
  return y
}
```

Provide a concise natural language description of the code using at most 218 characters.

*BLOOMZ:*

Generate a list of integers from a to b.

*Input (without previous input):*

Generate a list of integers from a to b.
Write functional code in JavaScript according to the description.

```
const generateIntegers = (a, b) => {
```

*BLOOMZ:*

```
  let integers = [];
  for (let i = a; i <= b; i++) {
    integers.push(i);
  }
  return integers;
}
```

Figure 35: **Incorrect solution from BLOOMZ to a HUMANEVALEXPLAIN JavaScript problem.**
BLOOMZ misses in its description that the function generates only the even digits between a and b, in ascending order. Thus, its synthesized function based on its own description simply collects all integers between a and b. BLOOMZ likely left out this part due to its bias for short and concise answers. However, this bias backfires for this task.

## R.5 STARCODER

*Input:*

```java
import java.util.*;
import java.lang.*;

class Solution {
    public Map<String, Integer> histogram(String test) {
        Map<String, Integer> dict1 = new HashMap<>();
        List<String> list1 = Arrays.asList(test.split(" " ));
        int t = 0;
        for (String i : list1) {
            if (Collections.frequency(list1, i) > t && !i.isEmpty()) {
                t = Collections.frequency(list1, i);
            }
        }
        if (t > 0) {
            for (String i : list1) {
                if (Collections.frequency(list1, i) == t) {
                    dict1.put(i, t);
                }
            }
        }
        return dict1;
    }
}
```

Provide a concise natural language description of the code using at most 423 characters.

*StarCoder:*

```java
public class Solution {
    public Map<String, Integer> histogram(String test) {

            int t = 0;

        }
    }
        if (t > 0) {
```

Write functional code in Java according to the description.

```java
import java.util.*;
import java.lang.*;

class Solution {
    public Map<String, Integer> histogram(String test) {
```

*StarCoder:*

```java
        Map<String, Integer> map = new HashMap<String, Integer>();
        int t = 0;
        for (int i = 0; i < test.length(); i++) {
            if (test.charAt(i) == 't') {
                t++;
            }
        }
        map.put("t", t);
        return map;
    }
}
```

Figure 36: **Incorrect solution from StarCoder to a HUMANEVALEXPLAIN Java problem.** Star-Coder is not capable of generating natural language descriptions and instead copies the solution. However, our post-processing removes substrings from the solution (§3) leading to a useless description. Given this description it generates an irrelevant function. As StarCoder cannot generate natural language explanations, we disqualify it from HUMANEVALEXPLAIN setting its score to 0 in Table 2.

## R.6   INSTRUCTCODET5+

*Input:*

Below is an instruction that describes a task. Write a response that appropriately completes the request.
### Instruction: Fix bugs in has_close_elements.

```python
from typing import List

def has_close_elements(numbers: List[float], threshold: float) -> bool:
    for idx, elem in enumerate(numbers):
        for idx2, elem2 in enumerate(numbers):
            if idx != idx2:
                distance = elem - elem2
                if distance < threshold:
                    return True

    return False

def check(has_close_elements):
    assert has_close_elements([1.0, 2.0, 3.9, 4.0, 5.0, 2.2], 0.3) == True
    assert has_close_elements([1.0, 2.0, 3.9, 4.0, 5.0, 2.2], 0.05) == False
    assert has_close_elements([1.0, 2.0, 5.9, 4.0, 5.0], 0.95) == True
    assert has_close_elements([1.0, 2.0, 5.9, 4.0, 5.0], 0.8) == False
    assert has_close_elements([1.0, 2.0, 3.0, 4.0, 5.0, 2.0], 0.1) == True
    assert has_close_elements([1.1, 2.2, 3.1, 4.1, 5.1], 1.0) == True
    assert has_close_elements([1.1, 2.2, 3.1, 4.1, 5.1], 0.5) == False

check(has_close_elements)
```

### Response:from typing import List

```python
def has_close_elements(numbers: List[float], threshold: float) -> bool:
```

*InstructCodeT5+:*

```python
    for idx, elem in enumerate(numbers):
        for idx2, elem2 in enumerate(numbers):
            if idx != idx2:
                distance = elem - elem2
                if distance < threshold:
                    return True

    return False
```

Figure 37: **Incorrect solution from InstructCodeT5+ to a HUMANEVALFIX Python problem.**
InstructCodeT5+ simply repeated the function without adding the missing "abs" statement to turn the distance into an absolute value. This is one of the simplest bugs that most other models are able to fix. For InstructCodeT5+ the prompt starts directly after "Response:" without a newline (Figure 20).

## R.7   STARCHAT-$\beta$

*Input:*

```c++
#include<stdio.h>
#include<math.h>
#include<vector>
#include<string>
#include<algorithm>
using namespace std;
#include<stdlib.h>
int specialFilter(vector<int> nums){
    int num=0;
    for (int i=0;i<nums.size();i++)
    if (nums[i]>10)
    {
        string w=to_string(nums[i]);
        if (w[0]%2==1 and w[w.length()-1]%2==1) num+=1;
    }
    return num;
}
```

Provide a concise natural language description of the code using at most 296 characters.

*StarChat-β:*

This code is a solution to the "Special Array" problem on LeetCode. It uses C++ and includes several standard libraries, including vector, string, and algorithm.
The code starts by declaring some variables and importing the necessary libraries. It then defines a specialFilter function that takes

*Input (without previous input):*

This code is a solution to the "Special Array" problem on LeetCode. It uses C++ and includes several standard libraries, including vector, string, and algorithm.
The code starts by declaring some variables and importing the necessary libraries. It then defines a specialFilter function that takes
Write functional code in C++ according to the description.

```c++
#include<stdio.h>
#include<math.h>
#include<vector>
#include<string>
#include<algorithm>
using namespace std;
#include<stdlib.h>
int specialFilter(vector<int> nums){
```

*StarChat-β:*

```c++
    int n = nums.size();
    int sum = 0;
    for(int i = 0; i < n; i++){
        sum += pow(nums[i], 2);
    }
    int target = (int)sqrt(sum);
    vector<int> res;
    for(int i = 0; i < n; i++){
        if(nums[i] > target){
            res.push_back(nums[i]);
        }
    }
    return res;
}
```

Figure 38: **Incorrect solution from StarChat-β to a HUMANEVALEXPLAIN C++ problem.** StarChat-β has a bias for excessively long answers. StarChat-β's description includes irrelevant parts like "..problem on LeetCode..includes several standard libraries.." (despite the problem not existing on LeetCode). As the description exceeds the 296-character limit for the code, it is cut off at the end.

## R.8 DIFF CODEGEN

*Input:*

```
<NME> truncate_number.py
<BEF> def truncate_number(number: float) -> float:
    """ Given a positive floating point number, it can be decomposed into
    and integer part (largest integer smaller than given number) and decimals
    (leftover part always smaller than 1).

    Return the decimal part of the number.
    >>> truncate_number(3.5)
    0.5
    """
    return number % 1.0 + 1.0
<MSG> Fix bugs in truncate_number.
<DFF>
```

*Diff Codegen 2B*

```
@@ -4,7 +4,7 @@ def truncate_number(number: float) -> float:
     and integer part (largest integer smaller than given number) and decimals
     (leftover part always smaller than 1).

-    Return the decimal part of the number.
+    Return
```

Figure 39: **Incorrect solution from Diff Codegen 2B to a HUMANEVALFIXDOCS Python problem.** Diff Codegen 2B suggests an irrelevant diff modifying parts of the docstring. The model commonly outputs diffs that modify the docstring or an import statement and rarely addresses the actual bug.

## S  LIMITATIONS AND FUTURE WORK

**Model Execution**   A promising avenue for improving performance on HUMANEVALFIX is letting the model execute the given code or its own generated code and inspect its output (Chen et al., 2022; 2023c; Yasunaga & Liang, 2021; Li et al., 2022a; Gao et al., 2023; Dong et al., 2023; Zhang et al., 2023c; Madaan et al., 2023b; Ni et al., 2023; Gou et al., 2023; Hu et al., 2023; Taylor et al., 2022; Nye et al., 2021). This could allow the model to discover which unit tests are failing and for what reason. The model could then simply iterate on the function until all unit tests are passing. We leave explorations of this strategy to improve performance on HUMANEVALPACK to future work.

**Multi-file changes**   For the creation of COMMITPACK, we have filtered out any commits that affect multiple files to ensure commits are very specific and account for the fact that most current models are only capable of operating on a single file. Allowing models to take multiple files as input and modify multiple files given a single instruction is a promising direction for future work. There is active research on using repository-level context (Ding et al., 2022; Shrivastava et al., 2023a;b; Zhang et al., 2023a; Liu et al., 2023d) and the necessary long context windows (Dai et al., 2019; Press et al., 2021; Sun et al., 2021; Dao et al., 2022; Peng et al., 2023; Liu et al., 2023c; Chen et al., 2023b).

**Length-awareness**   Current Code LLMs including OCTOCODER struggle with awareness about the length of their generated output. For HUMANEVALEXPLAIN, we instruct the models to limit their output to a given number of characters. While it is trivial for humans to count characters and adhere to the limit, all models tested frequently generate far too many characters. Prior work has shown that human raters are biased towards preferring longer texts (Wu & Aji, 2023) regardless of content. All models evaluated are instruction tuned on text that was at least indirectly assessed by human raters, hence they may be biased towards generating longer texts even if it means including literary bloat.

**Better evaluation**   Evaluating code instruction models is challenging for several reasons: **(1) Prompting:** The prompt can significantly impact the performance of large language mod-

els (Brown et al., 2020; Zhou et al., 2022; Muennighoff, 2022; Babe et al., 2023). To ensure fair evaluation we use the prompting format put forth by the respective authors of the models and a simple intuitive prompt for models without a canonical prompt (see Appendix Q). However, this may put models without a canonical prompt recommendation (e.g. BLOOMZ, GPT-4) at a slight disadvantage. OCTOCODER and OCTOGEEX perform best when prompted using the same format we use during training (Figure 18) and we recommend always using this format at inference.

**(2) Processing:** Models may accidentally impair otherwise correct code by e.g. including a natural language explanation in their output. We largely circumvent this issue through the use of strict stopping criteria and careful postprocessing (e.g. for GPT-4 we check if it has enclosed the code in backticks, and if so, extract only the inner part of the backticks discarding its explanations).

**(3) Execution:** When executing code to compute pass@k, it is important that the generated code matches the installed programming language version. Models may inadvertently use expressions from a different version (e.g. they may use the Python 2 syntax of `print "hi"`, which would fail in a Python 3 environment). In our evaluation, we did not find this to be a problem, however, as models become more capable, it may make sense to specify the version. Future prompts may include the version (e.g. "use JDK 1.18.0") or provide models with an execution environment that has the exact version installed that will be used for evaluation.

**(4) Comprehensiveness:** Executing code can only reflect functional correctness lacking a comprehensive understanding of quality. Compared to execution-based evaluation, the human judgment of code quality can be considered more comprehensive as humans can consider factors beyond correctness. Directly hiring human annotators can be inefficient and expensive, and therefore researchers have explored approaches to automate human-aligned evaluation via LLMs (Fu et al., 2023; Liu et al., 2023e; Zhuo, 2023). However, recent work (Wang et al., 2023b) suggests LLM-based evaluation can be biased towards certain contexts. Future work on automating the human-aligned evaluation of instruction tuned Code LLMs while avoiding such bias is needed.

**Reward Models** Our commit datasets, COMMITPACK and COMMITPACKFT, also lend themselves well for learning human preferences (Ethayarajh et al., 2024; Rafailov et al., 2024). The changed code after a commit generally represents a human-preferred version of the code (else the code would not have been modified). Thus, one could train a reward model that given the code before and after a commit, learns that the code afterward is better. Similar to prior work (Christiano et al., 2017; Ouyang et al., 2022), this reward model could then be used to guide a language model to generate code that is preferred by humans.

## T    VERSION CONTROL

**V1 → V2:**

- Added Appendix O on HUMANEVALEXPLAIN metrics
- Added Appendix N on Fill-in-the-Middle
- Expanded the motivation for limiting the number of tokens in COMMITPACKFT in Appendix E
- Fixed the StarCoder HUMANEVALFIXDOCS in Appendix K thanks to Abhijeet Awasthi
- Specified the programming languages of CommitPackFT (CommitPack has 350 languages while CommitPackFT has 277 after filtering)
- Made small writing improvements throughout

## U   OCTOBADPACK

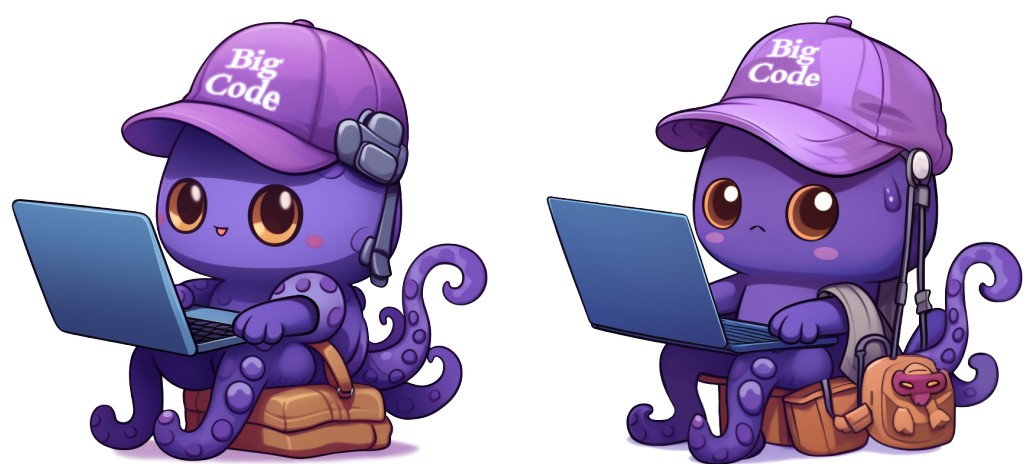

Figure 40: OCTOPACK (left) and her evil brother OCTOBADPACK (right).