# OpenReview forum: "OctoPack: Instruction Tuning Code Large Language Models"
_ICLR.cc/2024/Conference — ICLR 2024 spotlight_

### Official Review · Reviewer_eETk · 2023-10-31

**Soundness:** 3 good
**Presentation:** 3 good
**Contribution:** 3 good
**Rating:** 8
**Confidence:** 4

**Summary:**

The authors have compiled CommitPack, a 4TB dataset of permissively licensed git commits across 350 programming languages. They have strictly filtered CommitPack to derive clear imperative instructions, thus creating the instruction fine-tuning dataset, CommitPackFT.

To better evaluate instruction-tuned, large language models for code, they introduce HumanEvalPack. This extension of HumanEval encompasses three scenarios—code repair, code explanation, and code synthesis—and extends support to six programming languages.

They have conducted comparisons between different instruction tuning datasets using 5,000 random samples from CommitPackFT. The results demonstrate that combining CommitPackFT with OASST yields the best performance on HumanEvalPack.

They have released their datasets and the instruction-tuned code language models, OctoCoder and OctoCodeGeeX.

**Strengths:**

- By ensuring that the data collected and the models tuned are permissively licensed and free of proprietary data, the authors make a significant contribution to the open-source research community.
- They put a lot of effort into decontaminating the dataset and guaranteeing the soundness of HumanEvalPack (for example, by removing any solution overlap between the code and the generated explanation), which lends credibility to the results.
- The ablation studies and qualitative analysis yield many interesting conclusions and offer insights for further research.

**Weaknesses:**

- One thing I find unclear about the writing is the name of CommitPackFT.
As I understand, throughout the paper, the authors instruction-tuned the models using a 5000-sample subset of CommitPackFT, as mentioned on page 4
>For instruction tuning our models, we select 5,000 random samples from COMMITPACKFT across the 6 programming languages that we evaluate on.
Since you only used a subset of CommitPackFT to fine-tune the model, it might look a bit strange to call the entire dataset "FT".

**Questions:**

I wonder if instruction tuning the mode on more data from CommitPackFT can lead to even better performance.

---

> ### Author Response · Authors · 2023-11-16
>
> Thanks a lot for your review of the work.
>
>
> Weaknesses:
>
> We actually intended the FT to stand for `filtered`, thus referring to CommitPackFT being a filtered version of CommitPack. It is a nice coincidence that it could also refer to finetuning. We only used 5,000 samples, because similar to prior work (such as LIMA from Meta AI), we find that very few samples are needed for instruction tuning. Future work may consider finetuning on the entirety of CommitPackFT!
>
> Questions:
>
> As mentioned above, we found that the models converge extremely quickly. In fact, the 5,000 samples are not even entirely seen during our instruction tuning of OctoCoder. We think that this is due to most of the capabilities being learned during pre-training, while instruction tuning mostly serves the purpose of teaching the model the expected input and output format.

---

### Official Review · Reviewer_LDX4 · 2023-10-31

**Soundness:** 3 good
**Presentation:** 4 excellent
**Contribution:** 3 good
**Rating:** 8
**Confidence:** 5

**Summary:**

This work proposes to instruction-tune LLMs for code by using commit messages that clearly describe the change to the code in an imperative style, along with the original and new programs. It also introduces an extended version of HumanEval that spans 6 languages and three tasks as a benchmark. The results indicate that the proposed commit-based dataset in combination with OASST (from prior work) forms the strongest fine-tuning mixture, yielding strong results when fine-tuning StarCoder and CodeGeeX.

**Strengths:**

This work makes multiple substantial and useful contributions to the domain of code-specific instruction-tuning. As it notes, many successful recent approaches have been based on data from proprietary models. Using commit messages from permissive repositories offers a large dataset of relatively easy-to-acquire, and apparently moderately useful, data. The benchmark it contributes is also quite useful, moving substantially beyond the standard HumanEval set.

The work is quite well written and the supplementary material offers a very thorough view of the methods and results. Overall, this is a good paper.

**Weaknesses:**

The methodology makes a number of decisions that are not particularly well explained or defended. While none of these are critical, several of these would benefit from additional ablations and analyses, or an acknowledgement as a limitation. In no particular order:

The discussion around CommitPack leads with the rather enormous size of the initial dataset (~4TB). While this is accurate, the version used in this work is just 0.05% this size (2GB). This work offers no validation or experiments involving the original set. This both creates the impression that the 4TB set is likely to very noisy, and suggests that this work should really primarily emphasize the 2GB portion as far as concrete contributions of this paper are concerned. That would involve amending text like in the contributions on P2 and in the abstract, which don't mention the size of the subset that was actually used at all. This also relates to the "orders of magnitude more" comment on P9, which seems quite inconsequential.

Continuing the discussion of instruction data, it is not clear from the work why just 5K samples were chosen for fine-tuning. Was the concern that the model would drift too far if more samples were used? It is also not clear why (a) StarCoder was granted 3 extra samples, but that seems pretty inconsequential, and (b) OASST used nearly twice as many examples. Is that because the OASST samples are of higher quality? Was an ablation performed to choose these ratios? This is perhaps a particularly salient issue because one takeaway from this work is that CommitPackFT is not usable on its own. The experiments only ever report results of CommitPackFT + OASST (or similar combinations). This combination mostly boosts "Fix" performance; the results on the other two datasets are about even or worse than just using OASST by itself (Tab. 13). Perhaps the idea is that OASST is of very high quality but too small (are the ~8.5K samples used here the entire usable available subset?), so adding reasonably good samples increases performance? In any case, as is, the impression I get is that CommitPackFT isn't of particularly high-quality, but helps because it provides a bit more coding knowledge to OASST. More experiments could help contradict (or proof) that impression.

The work makes a rather strong argument, that building on HumanEval is a positive since it is so common that it is typically filtered from training data. That argument seems quite challenging to back up, given the absence of insight into how many LLMs are trained and anecdotal observations that pretrained models perform worse on new coding problems (e.g. from CodeContests). I would suggest a more moderated discussion of this choice, acknowledging the use of HumanEval as a potential limitation.

StarCoder is highlighted as performing very poorly in the explanation task, because it is unable to generate explanatory text. I wonder if this overlooks the obvious: query StarCoder to predict a docstring/javadoc/other comment using its FiM capabilities. The model has naturally not encountered samples were code is followed by a request for an explanation, but it would likely do relatively well when prompted to predict a comment above or at the start of a method. Although this somewhat stretches the definition of an "explanation", it does give the models more of a chance than the current setup.

Appendix C notes that inputs were filtered to just 768 tokens in length, for the complete before-code + message + after-code series. That is a very low limit by modern language modeling standards. Tab. 8 reinforces that this leads to very short programs compared to the natural distribution of commits. What was the motivation for these limits, and how does this impact the dataset's usefulness for downstream use-cases, where programs are rarely as short as those in the HumanEval suite?


Minor notes:

- The center panel in Fig. 3 might benefit from a visual improvement. Without reading the text, it was not clear to me what was happening here; as presented, it looks like the model is prompted with the code, predicts text, than is prompted with more text and predicts the code again, all in one input sample. Perhaps consider inserting a blank space between the two tasks (i.e., before the second model input) and moving the arrow to connect between the two halves.
- Sec. 3, P5: what experiments were conducted to confirm that GPT-4's accuracy only varies by 2% when sampling 1 sample vs. 20? I would imagine that the variation could be quite a bit more on some tasks. It might be worth sampling, say, 2-5 samples to balance cost and precision.
- It would probably help the work to also report BLEU/METEOR scores on the explanation generation task. This might offer a complementary view to the proposed automated metric, which is potentially somewhat lossy.
- A few odd notes from the appendix: why was "can’t you see i’m updating the time?" used as a prominent commit message filter (Tab. 5)? This doesn't seem to be a common phrase. Why was only data up till 2016 used? There are archives with more recent GitHub data.

**Questions:**

To avoid redundancy, please consider the questions raised in the weaknesses section above. Primarily, focus on the question of CommitPackFT's value given the relatively limited results (e.g. compared to OASST) and the various places where limitations may need to be acknowledged more clearly.

---

> ### Author Response · Authors · 2023-11-16
>
> Thanks a lot for your very thorough review.
>
> Weaknesses:
>
> 1. We do use the entirety of CommitPack for pretraining in `Appendix F`. We should have made this clearer, so we added a pointer to this appendix section in `Section 2. CommitPack: Code Instruction Data`. Given this, we would leave our claims as is, but please let us know if you would still adjust them.
> 2. The main reason was that we did not find many samples to be needed for instruction tuning similar to LIMA from Meta AI, where they only used 1000 samples. OctoCoder is only finetuned for 35 steps with a batch size of 32 and packing (Appendix L) corresponding to ~1100 - 8000 samples (depending on the packing efficiency). Regarding (a) and (b), we only made sure all datasets had the same order of magnitude, thus as our filtered OASST and StarCoder were already in that regime, we left them as is and only subsampled the much larger CommitPackFT and xP3x. As training does not cover the entirety of the dataset, we also don’t think that our filtered OASST is too small. Indeed CommitPackFT mainly boosts code fixing. One thing to note is that we only ablated on the Python split of HumanEvalPack. A key feature of CommitPackFT is its diversity across languages. We do not perform fine-tuning on CommitPackFT only, as it does not contain samples with natural language targets (See paragraph on `Importance of samples with natural language targets`).
> 3. Thanks for pointing this out. Indeed, inventing entirely new problems would have been better than building on HumanEval. We have rephrased the end of Section 3 to be more moderate and acknowledge this limitation.
> 4. This is a very interesting idea that we did not think of. We ran StarCoder with FIM on HumanEvalExplain and obtained the results below.
>
> | Model      | Python | JavaScript | Java | Go | C++ | Rust | Avg. |
> |------------|--------|------------|------|----|-----|------|------|
> | StarCoder  |   19.4   |   17.6     |  16.3   | 11.8  |  17.9  |  16.7   |   16.6  |
>
> We have observed that StarCoder demonstrates the ability to generate easily readable docstrings (e.g., ``Returns True if there are two elements in the list that are within the threshold’’)  when prompted with FIM. The corresponding average pass@1 results across different programming languages is approximately 16.6%. We have also added these results to the Appendix. Thanks a lot for your awesome suggestion!
>
> 5. Great point. The reason for this limit was that a large fraction of the code after is usually the same as the code before. Thus, much of the finetuning would be wasted on teaching the model to copy code from the before to the after. To increase the signal per token (and thus have the model learn faster) and avoid having a model that always repeats user input, we only finetune on relatively short commit pairs. We explain this in Appendix D - we have added some more details to that explanation.
>
> Minor notes:
> 1. Thanks, we have updated the plot according to your suggestion.
> 2. Good point, we have rephrased that sentence to `For GPT-4, we generate n=1 samples. Using n=1 instead of n=20 for GPT-4 only changed scores from 75.0% to 75.2% pass@1 on HumanEvalSynthesize Python while providing 20x cost savings.` which hopefully clarifies this?
> 3. As suggested, we have added a comparison to BLEU and METEOR in Appendix N. Note that for BLEU we assume that the docstring is a ground-truth explanation which can be a problematic assumption as docstrings are not necessarily meant to be explanations.
> 4. Surprisingly, “can’t you see I’m updating the time?” is one of the most common commit messages in our GitHub dump. You can check this blog post where it also appears as the 6th most common message: https://dev.classmethod.jp/articles/my-favorite-bigquery-dataset-newshiro/ or this one from Google: https://codelabs.developers.google.com/codelabs/cloud-bigquery-csharp#5. We’re not sure why this is the case, but decided to remove those samples, as it is not a useful commit message. The data we used unfortunately only contained data until 2016 (https://github.blog/2016-06-29-making-open-source-data-more-available/). We are working on releasing a new commit dataset with more recent data.

---

> > ### Comment · Reviewer_LDX4 · 2023-11-22
> >
> > Thanks for your clarifications! I appreciated the added results & text. To respond to point 1: that's good to know. I would still encourage you to make mention of the 2GB number somewhere in the introduction just to set expectations, e.g., add that there is a curated/cleaned subset of 2GB at the end of the first contribution (on P2).
> >
> > Otherwise everything looks good now. I'm naturally still quite positive about the work and support its acceptance.

---

> > > ### Author Response · Authors · 2023-11-22
> > >
> > > Thank you. As suggested we have added the 2GB number at the end of the first contribution as follows:
> > >
> > > `4TB of permissively licensed code commits across 350 programming languages for pretraining and a filtered 2GB variant containing high-quality code instructions used for finetuning`
> > >
> > > Let us know if there is anything else.

---

### Official Review · Reviewer_SuJc · 2023-11-01

**Soundness:** 3 good
**Presentation:** 3 good
**Contribution:** 3 good
**Rating:** 6
**Confidence:** 3

**Summary:**

This paper studies the training and evaluation of Code LLMs with instructions. It first creates COMMITPACKFT, 2GB of high-quality code with commit messages that assimilate instructions. Then it constructs HUMANEVALPACK, a human-written benchmark covering 3 different tasks for 6 programming languages. Afterward, this work ablates several instruction datasets and ﬁnds out that COMMITPACKFT combined with natural language data leads to the best performance. Moreover, the models, OCTOCODER and OCTOGEEX, outperform GPT-4.

**Strengths:**

1. The new proposed framework that enhances the instruction tuning on LLMs is impressive.
2. The paper is well-organized and well-written with clear motivations, detailed discussion, nice figures, and sufficient comparison experiments, making it easy to follow and understand.
3. This work performs comprehensive experiments over benchmark data to show the effectiveness in several settings.
4. This work creates new datasets, i.e., COMMITPACK and COMMITPACKFT which are 4TB of permissively licensed code commits across 350 programming languages for pretraining and a ﬁltered variant containing high-quality code instructions. These datasets will make contributions to this field in future research.

**Weaknesses:**

1. This work finds out that OCTOCODER and OCTOGEEX, perform best among permissive models. I am curious about the reason behind that. Besides, a detailed introduction about OCTOCODER and OCTOGEEX would help understand the experimental results.
2. Still in experimental discussion, I am also curious about the performance of OCTOCODER and OCTOGEEX over the HUMANEVALEXPLAIN dataset. Why does OCTOGEEX have better performance in Go and Rust compared to  OCTOCODER? What are the differences between OCTOCODER and OCTOGEEX?

**Questions:**

1. Why do OCTOCODER and OCTOGEEX perform best among permissive models?
2. What are the differences between OCTOCODER and OCTOGEEX?
3. Why does OCTOGEEX have better performance in Go and Rust languages compared to  OCTOCODER?

---

> ### Author Response · Authors · 2023-11-16
>
> Thanks a lot for your review. We would like to point out that OctoCoder and OctoGeeX do not outperform GPT-4.
>
> Weaknesses:
> 1. Thanks for noting this! We have added a reference to Appendix L at the end of Section 4.1, where the models (base model, hyperparameters, etc.) are better explained.
> 2. The difference is the pre-trained base model and some minor hyperparameters. While OctoCoder is initialized from StarCoder, OctoGeeX is initialized from CodeGeeX2. There are tiny differences in the hyperparameters during finetuning, which are detailed in `Appendix O: Hyperparameters`. Indeed OctoGeeX outperforms OctoCoder on Go and Rust for HumanEvalExplain. The base model (CodeGeeX2) also performs better than OctoCoder’s base model (StarCoder) for Go and Rust (https://github.com/THUDM/CodeGeeX2/blob/main/README_EN.md) which could be the reason for this.
>
>
> Questions:
> 1. See Weaknesses 1.
> 2. See Weaknesses 1.
> 3. See Weaknesses 2.
>
> Let us know if we can elaborate further on any part.

---

### Author Response · Authors · 2023-11-16

We thank all reviewers for their detailed reviews. We have made the following updates to the paper:

1. Added a reference to Appendix L at the end of Section 4.1
2. Added a reference to Appendix F in Section 2
3. Added Appendix N on HumanEvalExplain metrics (pass@1 vs BLEU vs METEOR)
4. Added Appendix M on using Fill-in-the-Middle for HumanEvalExplain with StarCoder
5. Expanded the motivation for limiting the number of tokens in CommitPackFT in Appendix D
6. Small writing improvements throughout
7. Added a Version Control Appendix S which specifies all the updates

We appreciate the positive sentiment expressed about the OctoPack resources and are very excited about future work building on CommitPack, HumanEvalPack & OctoCoder/GeeX!

---

### Meta-Review · Area_Chair_axBD · 2023-12-08

**Metareview:**

This is a dataset and evaluation benchmark that should be valuable to the community. All reviewers voted to accept.

The main problem mentioned is that is subsampling 4TB of commit data is misleading since only 2GB filtered is used. The authors clarified this in the intro but the 4TB number still makes the headline in the abstract. A small 5K sample was used for instruction fine-tuning in the model, raising some questions about instruction tuning code models.

There is also no comparisons to more recent models based on Code Llama that does better among open models. While some headline accuracy results may no longer hold in this fast moving area, the dataset is remains valuable. The authors are encourages to focus on that and in establishing the quality and specifications of the dataset.

**Justification For Why Not Higher Score:**

the main contributions can be communicated in a spotlight.

**Justification For Why Not Lower Score:**

N/A

---

### Decision · Program_Chairs · 2024-01-16

Accept (spotlight)